# Immunoglobulin G *N*-glycan markers of accelerated biological aging during chronic HIV infection

Leila B. Giron [1], Qin Liu [1], Opeyemi S. Adeniji[1], Xiangfan Yin[1], Toshitha Kannan [1], Jianyi Ding[1], David Y. Lu[1,2], Susan Langan[3], Jinbing Zhang[3], Joao L. L. C. Azevedo[1], Shuk Hang Li [4], Sergei Shalygin [5], Parastoo Azadi[5], David B. Hanna[6], Igho Ofotokun[7], Jason Lazar[8], Margaret A. Fischl[9], Sabina Haberlen[3], Bernard Macatangay[10], Adaora A. Adimora[11], Beth D. Jamieson [12], Charles Rinaldo[10], Daniel Merenstein[13], Nadia R. Roan [14,15], Olaf Kutsch[16], Stephen Gange [3], Steven M. Wolinsky[17], Mallory D. Witt[18], Wendy S. Post[3], Andrew Kossenkov[1], Alan L. Landay[19], Ian Frank[4], Phyllis C. Tien[15], Robert Gross[4], Todd T. Brown[3] & Mohamed Abdel-Mohsen [1] ✉

People living with HIV (PLWH) experience increased vulnerability to premature aging and inflammation-associated comorbidities, even when HIV replication is suppressed by antiretroviral therapy (ART). However, the factors associated with this vulnerability remain uncertain. In the general population, alterations in the *N*-glycans on IgGs trigger inflammation and precede the onset of aging-associated diseases. Here, we investigate the IgG *N*-glycans in cross-sectional and longitudinal samples from 1214 women and men, living with and without HIV. PLWH exhibit an accelerated accumulation of pro-aging-associated glycan alterations and heightened expression of senescence-associated glycan-degrading enzymes compared to controls. These alterations correlate with elevated markers of inflammation and the severity of comorbidities, potentially preceding the development of such comorbidities. Mechanistically, HIV-specific antibodies glycoengineered with these alterations exhibit a reduced ability to elicit anti-HIV Fc-mediated immune activities. These findings hold potential for the development of biomarkers and tools to identify and prevent premature aging and comorbidities in PLWH.

Even with long-term suppressive antiretroviral therapy (ART), people living with chronic HIV infection (PLWH) prematurely experience a high incidence of aging-associated diseases, including cardiovascular disease (CVD), cancers, and neurocognitive disorders[1]. There are considerable gaps in our understanding of the pathophysiological mechanisms driving the development of such comorbidities in PLWH; however, many of these comorbidities are linked to a chronic inflammatory state called inflammaging, commonly observed in elderly individuals[2–4]. The precise mechanisms driving inflammaging in PLWH are not fully understood, but they may involve ongoing HIV production, cytomegalovirus (CMV) infection, loss of regulatory T cells, microbial translocation, and other undetermined host and viral factors[5]. Comprehensive understanding of the factors associated with inflammaging in PLWH can facilitate the development of biomarkers to predict the occurrence or severity of inflammaging-associated comorbidities, and may aid in the development of tools to prevent the onset of these comorbidities.

Aberrant host glycosylation has recently emerged as a key driver of chronic inflammation and accelerated biological aging in the general population[6-9]. Within the circulating glycome, the glycans on antibodies are especially critical as these are linked to systemic and chronic inflammatory responses. Specifically, the glycans on the Fc domain of circulating immunoglobulins G (IgGs) play a crucial role in regulating antibody non-neutralizing functions, including antibody-dependent cell-mediated cytotoxicity (ADCC), antibody-dependent cellular phagocytosis (ADCP), complement-dependent cytotoxicity (ADCD), and various pro- and anti-inflammatory activities[10,11]. Extensive literature shows that IgG N-glycan alterations can mechanistically influence inflammation and associated comorbidities. For example: (1) Sialic acid: one of the most studied examples is intravenous immunoglobulins (IVIGs), whose anti-inflammatory activity is dependent on the sialic acid moiety carried on their Fc[12]. Beyond IVIGs, the loss of IgG sialic acid plays a mechanistic role in the development of obesity-induced hypertension, and supplementation of sialic acid reduces obesity-induced hypertension in mouse models[13]. (2) Galactose: galactose on IgG leads to anti-inflammatory cascades by facilitating the interaction between CD32b and dectin-1 in myeloid cells[14]. (3) Fucose: afucosylated IgG glycans contribute to inflammation during SARS-CoV-2 infection[15].

Beyond inflammation, several studies have also demonstrated that alterations to IgG glycosylation are strongly associated with both chronological and biological aging in the general population. These alterations might even be better in predicting accelerated biological aging than traditional markers like shorter telomere length[16-20]. Such glycan traits are significantly altered in individuals with age-related illnesses, such as cardiovascular disease (CVD) and cancer[21-23]. Whether IgG glycosylation drives, or is simply a biomarker of aging, aging-related diseases, and aging-related comorbidities is still a subject of debate. However, changes in IgG glycosylation have been observed years before the onset of some diseases, indicating a potential causative role[24].

Despite the growing body of evidence suggesting a connection between an altered circulating IgG N-glyans and accelerated biological aging, it remains unclear whether chronic HIV infection accelerates the pace of aging-associated IgG N-glycan alterations. Our group previously showed that IgGs of PLWH had lower levels of circulating glycans suggested to be anti-inflammatory and anti-aging (i.e., sialylated and galactosylated glycans) than did IgGs of people living without HIV (PLWoH)[25]. However, in that prior study, the sample size was small, and the groups were not matched for demographic factors such as age, sex, and ethnicity. In addition, that study did not examine the potential upstream mechanisms of these alterations or the potential downstream consequences of them.

In the current study, we aim to address these limitations by investigating IgG N-glycans in both cross-sectional and longitudinal samples from women and men, living with and without HIV. We find that PLWH exhibit an accelerated accumulation of pro-aging-associated glycan alterations and heightened expression of senescence-associated glycan-degrading enzymes, compared to controls. These alterations correlate with elevated markers of inflammation and the severity of comorbidities, potentially preceding the development of such comorbidities. Mechanistically, HIV-specific antibodies glycoengineered with these alterations show a reduced ability to elicit anti-HIV Fc-mediated immune activities, potentially explaining their association with worse clinical outcomes.

## Results

### Long-term, ART-suppressed HIV infection is associated with sex-dependent IgG N-glycan alterations

We first sought to examine whether living with well-suppressed HIV infection is associated with alterations in IgG N-glycans. To isolate the influence of ART-suppressed HIV infection on IgG N-glycans, while minimizing confounding factors and selection biases, we selected cross-sectional samples from PLWH on suppressive ART and PLWoH counterparts enrolled in the MACS/WIHS Combined Cohort Study (MWCCS). The key feature of this cohort is the comparable recruitment, data collection procedures, and long-term retention of PLWH and well-matched PLWoH controls[26-29]. As it is documented that sex impacts IgG glycosylation[30], we separated our analysis by sex assigned at birth, specifically biologically-born women (hereafter referred to as women) and biologically-born men (hereafter referred to as men). We analyzed the IgG N-glycans of 254 women living with HIV (WLWH) on suppressive ART for at least five years with undetectable HIV viral load, weight less than 300 pounds, and with a median CD4 T cell count of 726 cells/mm³. We matched these samples with samples from 235 women living without HIV (WLWoH) matched to the WLWH in terms of age, race, and body mass index (BMI) (Table 1, Fig. 1a). To ensure a wide age range for analysis, we aimed to have a similar number of individuals in each of the following age categories: ≤45, 46-50, 51-55, 56-60, 60-65, and >65 years (Table 1, Fig. 1a). Using the same criteria, we also analyzed samples from 243 men living with HIV (MLWH) who were on suppressive ART with a median CD4 T cell count of 698 cells/mm³ and 253 men living without HIV (MLWoH) matched controls (Table 1, Fig. 1a).

We identified 20 individual N-glycan structures within the bulk IgG N-glycans of these samples (Supplementary Fig. 1a). These glycan structures were grouped into five IgG glycan groups, depending on the presence or absence of four key monosaccharides: sialic acid (sialylated), galactose (agalactosylated, and terminal galactosylated), fucose (fucosylated), and bisecting GlcNAc (bisected) (Supplementary Fig. 1b).

We first assessed whether ART-suppressed HIV infection is associated with alterations in the levels of any of these IgG glycan groups, compared to controls, and whether the degree of alteration differed between WLWH and MLWH. We found that ART-suppressed HIV infection was associated with higher levels of agalactosylated glycans (pro-aging) in both men and women (FDR < 0.05; Fig. 1b). To examine whether this induction of agalactosylated glycans differed between women and men, we needed to account for potential confounding factors such as ethnicity and BMI, which differed between the women and men (Table 1). We employed multi-variable models that adjusted for age, ethnicity, and BMI. The interaction comparison revealed no significant differences in the association between HIV and IgG agalactosylated glycans between women and men (Fig. 1b). These findings suggest that living with HIV and/or ART use is associated with a similar induction in the levels of agalactosylated glycans (pro-aging) in WLWH and MLWH compared to their counterparts. Using a similar analysis approach, we examined the effects of HIV/ART on the other IgG glycan groups. We found that HIV/ART was associated with: (1) a reduction of terminally galactosylated glycans (anti-aging) in both men and women (Fig. 1c), (2) a reduction of sialylated glycans (associated with anti-inflammatory activities) in men but not women (Fig. 1d), (3) a reduction of core fucosylated glycans (which are involved in the modulation of IgG-mediated ADCC), in both men and women, but more profoundly in men than women (Fig. 1e), and (4) an induction of the bisected GlcNAc glycans (pro-aging) in both men and women (Fig. 1f), compared to controls. A detailed comparison of the 20 individual glycan structures (Supplementary Table 1) supported the group analysis that living with HIV and/or ART use are associated with an alteration in the levels of several IgG N-glycan structures.

Because previous studies in the general population indicated that menopause and sex hormones have an impact on IgG glycosylation[30-33], we performed a secondary analysis of the women samples based on self-reported menopause status: pre-menopause (n = 60 WLWoH and n = 35 WLWH), peri-menopause (n = 21 WLWoH and n = 20 WLWH), or post-menopause (n = 152 WLWoH and n = 198 WLWH). We compared the IgG N-glycan groups (Supplementary Fig. 2)

**Table 1 | The characteristics of participants in studies depicted in Figs. 1–4 and 8**

| | | Women | | | | Men | | | |
|---|---|---|---|---|---|---|---|---|---|
| | | PLWoH (*n* = 235) | | PLWH on ART (*n* = 254) | | PLWoH (*n* = 253) | | PLWH on ART (*n* = 243) | |
| | | *n* (%) | Median, IQR | *n* (%) | Median, IQR | *n* (%) | Median, IQR | *n* (%) | Median, IQR |
| Age (years) | ≤45 | 33 (14.0%) | 43 (5) | 23 (9.1%) | 44 (4) | 17 (6.7%) | 44 (2) | 20 (8.2%) | 44 (2.5) |
| | 46–50 | 56 (23.8%) | 48 (2) | 50 (19.7%) | 48 (2) | 36 (14.2%) | 48 (3) | 43 (17.7%) | 47 (3) |
| | 51–55 | 55 (23.4%) | 53 (2) | 68 (26.8%) | 53 (2) | 54 (21.3%) | 53 (3) | 58 (23.9%) | 53 (2) |
| | 56–60 | 50 (21.3%) | 58 (3) | 59 (23.2%) | 58 (3) | 63 (24.9%) | 58 (3) | 54 (22.2%) | 58 (2) |
| | 61–65 | 35 (14.9%) | 63 (3) | 45 (17.7%) | 62 (3) | 49 (19.4%) | 62 (3) | 36 (14.8%) | 63 (2) |
| | >66 | 6 (2.6%) | 69 (6) | 9 (3.5%) | 71 (6) | 34 (13.4%) | 69 (4.5) | 32 (13.2%) | 68.5 (4) |
| Race | White, non-Hispanic | 22 (9.4%) | – | 30 (11.8%) | – | 166 (65.6%) | – | 128 (52.7%) | – |
| | White, Hispanic | 12 (5.1%) | – | 19 (7.5%) | – | 11 (4.3%) | – | 14 (5.8%) | – |
| | Black, non-Hispanic | 175 (74.5%) | – | 169 (66.5%) | – | 68 (26.9%) | – | 78 (32.1%) | – |
| | Black, Hispanic | 4 (1.7%) | – | 5 (2%) | – | 2 (0.8%) | – | 4 (1.6%) | – |
| | American Indian or Alaskan Native | 1 (0.4%) | – | 1 (0.4%) | – | 1 (0.4%) | – | 1 (0.4%) | – |
| | Asian or Pacific Islander | 0 (0%) | – | 0 (0%) | – | 1 (0.4%) | – | 1 (0.4%) | – |
| | Other | 1 (0.4%) | – | 1 (0.4%) | – | 3 (1.2%) | – | 2 (0.8%) | – |
| | Other Hispanic | 20 (8.5%) | – | 29 (11.4%) | – | 1 (0.4%) | – | 15 (6.2%) | – |
| BMI | | | 30.5 (9.7) | | 30.3 (9.5) | | 26.2 (5.25) | | 25.5 (5.85) |
| Diabetes | No | 162 (68.9%) | – | 181 (71.3%) | – | 196 (77.5%) | – | 191 (78.6%) | – |
| | Yes | 73 (31.1%) | – | 73 (28.7%) | – | 30 (11.9%) | – | 34 (14%) | – |
| | Insufficient data | 0 (0%) | – | 0 (0%) | – | 27 (10.7%) | – | 18 (7.4%) | – |
| Blood CD4 count (cells/mm³) | | – | – | – | 725.5 (381.75) | – | – | | 698 (364) |
| Plasma viral load (copies/ml) | | – | – | 254 (100%) | <50 | – | – | 243 (100%) | <50 |
| Blood nadir CD4 count (cells/mm³) | | – | – | – | 216 (213.75) | – | – | – | 306.5 (234) |

or individual glycan traits (Supplementary Table 2) among these menopause groups. Our findings confirmed that menopause and/or age are associated with a reduction in glycan structures that are considered anti-aging and anti-inflammatory (galactosylated and sialylated) and an increase in glycan structures that are considered pro-aging and pro-inflammatory (agalactosylated and bisected), in both WLWH and their WLWoH counterparts.

Together, these findings suggest that living with HIV is associated with distinct and likely detrimental IgG *N*-glycan alterations. These alterations include an accumulation of pro-aging glycans (agalactosylated and bisected GlcNAc), a decline in anti-aging glycans (terminally galactosylated and sialylated), and a decrease in glycans involved in the modulation of ADCC (fucosylated), with variations observed based on sex (Fig. 1g).

**IgG *N*-glycan alterations associated with living with HIV and/or ART use correlate with higher systemic markers of inflammation**
IgG N-glycan alterations have been linked to inflammatory activities[12,14,34–40]. Considering that ART-suppressed HIV infection significantly associates with alterations in IgG *N*-glycans, our next objective was to investigate whether the changes in IgG *N*-glycans associated with HIV/ART are correlated with inflammation. To achieve this, we selected 400 individuals, with 100 from each of the four groups in Fig. 1a. These were matched for age and ethnicity between groups of women and men. We measured the levels of 22 inflammatory markers known to play crucial roles in HIV pathogenesis (such as sCD14, sCD163, IL-6)[5], as well as markers associated with inflammatory aging[41]. Among the markers associated with inflammatory aging, we included three markers (CXCL9, Eotaxin, and leptin), which had been recently incorporated into an inflammatory aging clock capable of predicting

accelerated biological aging in the general population[41]. We found that, in general, PLWH exhibit higher levels of plasma inflammatory markers in a sex-dependent manner. For instance, IP-10 and sCD14 were elevated in women and men living with HIV compared to their controls. CXCL9 was higher in WLWH but not in MLWH compared to their respective controls. Additionally, levels of sCD163, IL-4, IL-5, IL-12p70, MIP-1α, and MCP-2 were elevated in MLWH but not in WLWH compared to their controls (Fig. 2a–i). We also investigated if these markers differed among women based on menopause status. Menopause was associated with increased levels of CXCL9 and Eotaxin, and the association of HIV infection on CXCL9 and IP-10 was more pronounced in post-menopausal women compared to pre-menopausal women (Supplementary Fig. 3).

Given that both IgG glycans and markers of inflammation differed by HIV serostatus, we then examined correlations. As depicted in Fig. 2j, we found negative correlations between the glycans that are depleted during chronic HIV infection (sialylated, galactosylated, and fucosylated glycans) and multiple markers of inflammation. Conversely, we found positive correlations between the glycans that are enriched during chronic HIV infection (agalactosylated and bisected GlcNAc glycans) and markers of inflammation. These data suggest that HIV-associated IgG *N*-glycan alterations are linked to systemic markers of inflammation, aligning with their known functions.

**The links between IgG *N*-glycans and chronological age are altered in PLWH**
Previous studies in the general population indicated that specific IgG *N*-glycans correlate with chronological age[16]. However, given that ART-suppressed HIV infection is associated with changes in the IgG *N*-glycans, we next sought to determine whether these changes altered the

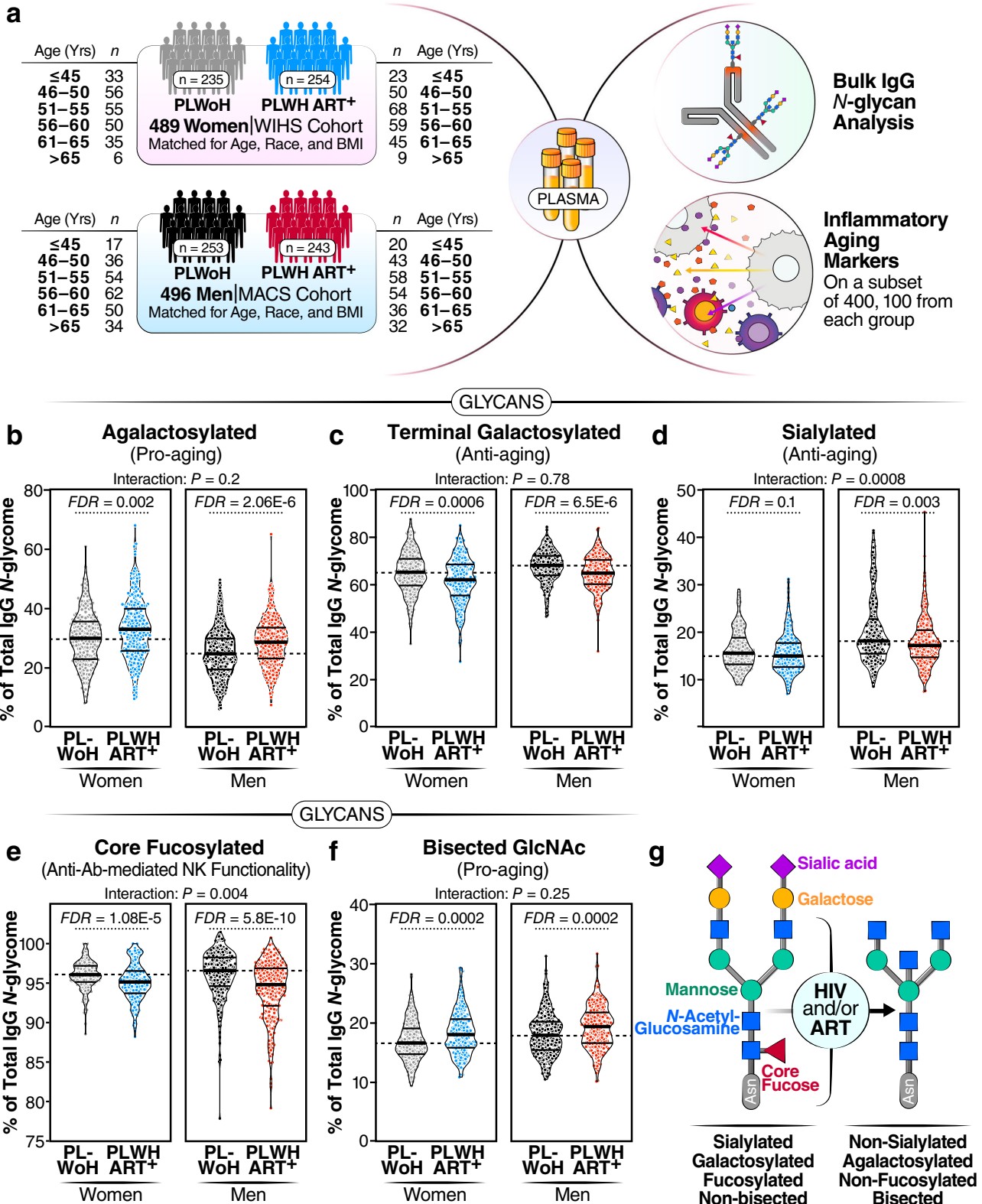

**Fig. 1 | Long-term ART-suppressed HIV infection is associated with sex-dependent IgG *N*-glycan alterations. a** Overview of the main study design. **b**–**f** Violin plots displaying the levels of IgG *N*-glycan groups in WLWH and MLWH undergoing long-term ART compared to their PLWoH controls. The median and interquartile range (IQR) are shown. Two-tailed unpaired *t* tests and false discovery rates (FDRs) were calculated to account for multiple tests over studied markers. Interaction *P* values were calculated using multivariable models, adjusting for age, ethnicity, and BMI. *N* = 985 biological samples. **g** A schematic summary illustrating the IgG *N*-glycan alterations associated with HIV infection and/or ART treatment. Source data are provided as a Source Data file.

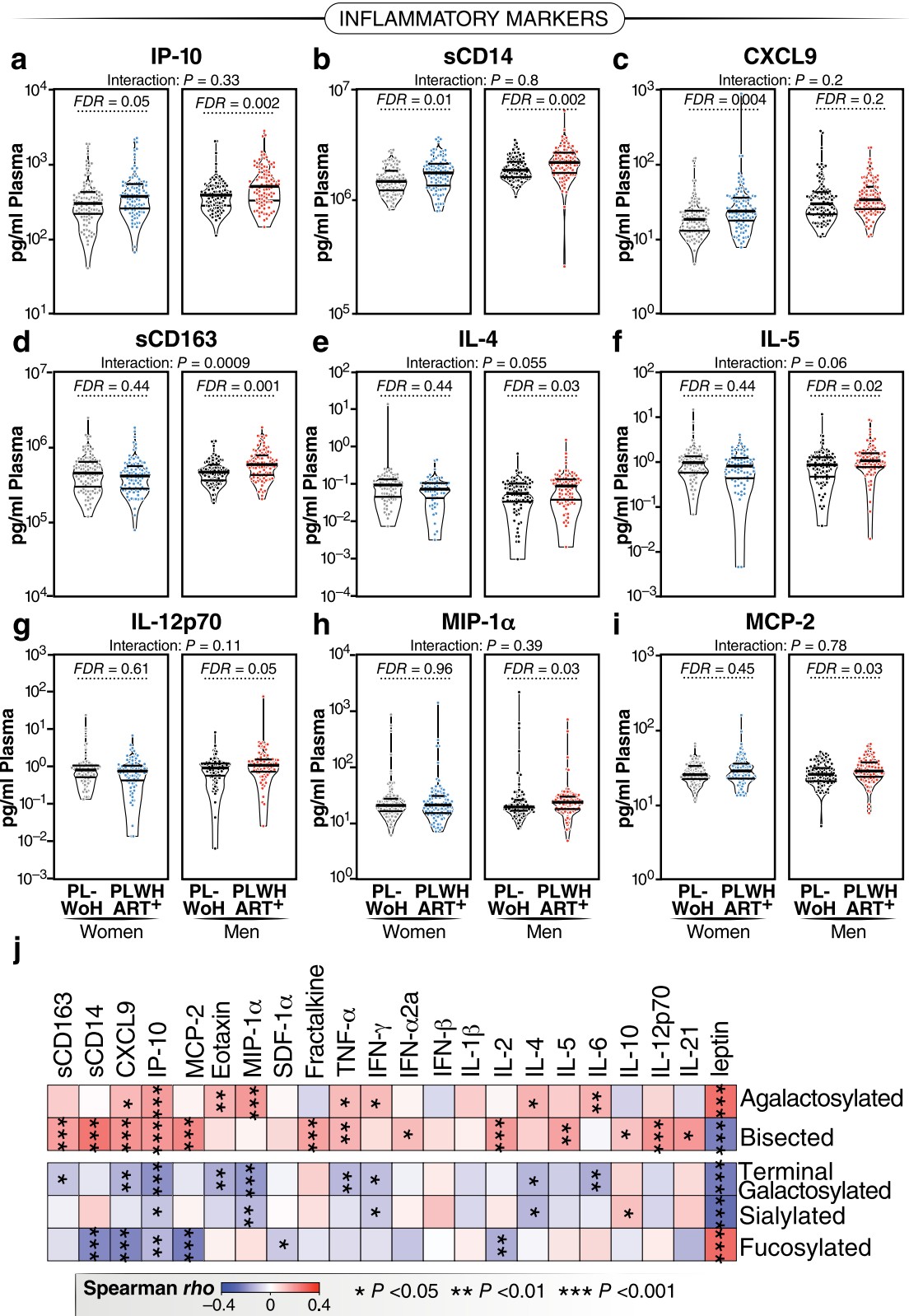

**Fig. 2 | HIV/ART-associated IgG *N*-glycan alterations correlate with higher markers of inflammation. a–i** Violin plots illustrating the levels of various plasma markers of inflammation in WLWH and MLWH undergoing long-term suppressive ART compared to PLWoH controls. Median and IQR are displayed. Two-tailed unpaired *t* tests and FDRs were employed to address multiple comparisons. Interaction *P* values were computed using multivariable models, adjusting for age, ethnicity, and BMI. *N* = 400 biological samples. **j** Two-tailed Spearman's rank correlation heatmap revealing the associations between IgG *N*-glycan groups (rows) and several plasma markers of inflammation (columns). *N* = 400 biological samples. Positive correlations are depicted in red, while negative correlations are shown in blue. \**P* < 0.05, \*\**P* < 0.01, and \*\*\**P* < 0.001. Source data are provided as a Source Data file.

relationship between IgG N-glycans and chronological age in PLWH. We identified three distinct patterns. First, several glycan structures exhibit a positive correlation with age in both PLWH and PLWoH. However, the slope of these correlations is significantly different in PLWH, suggesting that HIV either accelerates or decelerates the accumulation of these glycan structures over time. For instance, the agalactosylated A2 glycan trait (Fig. 3a) and the bisected GlcNAc grouped glycans (Fig. 3b) both positively correlated with age in PLWH and PLWoH, but the slope of these correlations differs significantly between the two groups ($P = 0.0075$ and $P = 0.059$, respectively). Second, specific glycan traits, such as FA2BG2 (Fig. 3c), do not correlate with age in one of the two populations but do correlate in the other population. Third, specific glycans such as FA2G2 (Fig. 3d) show a consistent correlation with age in both PLWH and PLWoH. The complete dataset can be found in Supplementary Table 3.

To validate these cross-sectional findings, we conducted an additional analysis focusing on the longitudinal changes in IgG N-glycan patterns from 23 WLWH, 24 MLWH, and 47 PLWoH. For this analysis, we used samples that have been collected from each participant at intervals of one or two years (from age ~50 to ~60 years old; Supplementary Table 4), resulting in a total of 6–7 samples per person. All participants were well-matched in terms of age, race, and BMI at the baseline visit. Using these data, we confirmed several findings from our cross-sectional analysis, despite the shorter age span of the longitudinal analysis. For instance, bisected GlcNAc glycans correlated positively with age in both PLWH and PLWoH over time, although the slope of the correlation differed between the two groups; FA2G2 glycan trait correlated negatively with age in both groups; and agalactosylated glycans and the FA2B glycan trait correlated positively with age, but the slope varied between the two groups (Supplementary Fig. 4). These findings suggest that living with HIV and/or ART use can alter the pace of age-associated IgG N-glycan alterations.

## Machine learning models based on IgG N-glycans indicate an acceleration of biological aging in PLWH

IgG glycans have been used as biomarkers for biological aging in the general population[16,42]. We, therefore, hypothesized that these IgG N-glycans could be used to estimate the rate of accelerated biological aging during well-suppressed HIV infection. Prior studies had employed a glycan-based clock consisting of multiple individual glycan features. However, those models were trained on samples from individuals who were not matched for ethnicity and demographic factors to PLWH, and thus cannot be directly employed to calculate the biological age of PLWH. To address this, we leveraged the analysis of IgG N-glycans in PLWoH from the MWCCS who serve as a comparison group to PLWH[26–29]. Using this data, we developed glycan-based machine learning models that can estimate the acceleration of biological age in PLWH.

We first used the "least absolute shrinkage and selection operator" (LASSO) model to identify the fewest number of individual IgG N-glycan traits that, when combined, correlated with chronological age in MLWoH with an average of zero difference between predicted age and chronological age. This would enable us to calculate the additional rate of acceleration in biological aging in MLWH that is specifically influenced by HIV and/or ART. Among the models evaluated (Supplementary Table 5), the best-performing one incorporated four glycan structures: A2, FA2B, FA2BG1, and A2G2S2. When this model was applied to data from MLWH, the results (as depicted in Fig. 4a: left) indicated an average acceleration of 3.52 years with a standard deviation of 7.47 years ($P < 0.0001$) between predicted age and chronological age, compared to their matched MLWoH. As recent studies suggest that the identification of the A2G2S2 glycan trait in isolated IgGs could be a result of transferrin contamination[43], and despite the high purity of our isolated IgG preparations, as depicted in Supplementary Fig. 1c–e, we evaluated a model excluding the A2G2S2

glycan traits and only including the A2, FA2B, and FA2BG1 glycan traits (Fig. 4b). This model mimicked our observation in the first model, showing an average of 3.72 years difference between predicted age and chronological age in MLWH, compared to their matched MLWoH.

As plasma inflammatory markers have also been employed as indicators of biological aging[41], we investigated whether models based on some of these markers could also estimate the rate of accelerated biological aging in MLWH. We also explored whether the glycan model's efficacy could be enhanced by combining plasma inflammatory markers with the four identified IgG glycan structures. First, we examined the correlation between inflammatory markers and chronological age in MLWH or WLWH (Supplementary Table 6). We found no inflammatory marker correlated with chronological age (FDR < 0.05) in MLWH, whereas three markers (CXCL9, Eotaxin, and TNF-α) did correlate with chronological age (FDR < 0.05) in WLWH. Next, we assessed whether these three inflammatory markers could be combined in a model. Both CXCL9 and Eotaxin exhibited unique correlations with age; however, their combination was insufficient to estimate the rate of biological aging in MLWH (Fig. 4a, b: middle; Supplementary Table 5). Furthermore, combining these two inflammatory markers with the four or three IgG N-glycans identified from the glycan models did not improve the predictive capacity of the pure glycan model (Fig. 4a, b: right; Supplementary Table 5).

Following a similar analysis approach, we identified two glycans (FA2G2S1 and FA2G2) in women that correlated with chronological age in WLWoH with an average of zero difference between predicted age and chronological age. However, a model based on these two glycans was insufficient in estimating the rate of biological age in WLWH (Fig. 4c: left; Supplementary Table 5). A combination of CXCL9 and Eotaxin moderately estimated the rate of biological aging in WLWH (Fig. 4c: middle; Supplementary Table 5). Notably, the best model for estimating the rate of biological aging in WLWH encompassed two glycans (FA2G2S1 and FA2G2) and two inflammatory markers (Fig. 4c: right; $P < 0.0001$; Supplementary Table 5). Applying this model to data from WLWH revealed an average acceleration of 2.95 years with a standard deviation of 10.63 years ($P = 0.0066$) between predicted and chronological age in WLWH.

Taken together, these findings suggest that IgG N-glycans can be employed in models to calculate biological aging in PLWH. However, the model requirements differ between MLWH and WLWH, whereby glycans alone suffice to estimate the rate of biological aging in MLWH, while a combination of glycans and inflammatory aging markers is necessary for similarly robust estimation in WLWH.

## IgG agalactosylation and bisecting GlcNAc correlate with the development and severity of coronary atherosclerosis in PLWH

We next examined whether the IgG N-glycan dysregulations associated with living with chronic HIV are also associated with the development of age- and inflammation-related comorbidities in PLWH. To accomplish this, we designed a case-control study within the MACS cohort[44,45]. The cases were individuals with coronary artery stenosis of ≥50% in one or more coronary segments, whereas the controls were individuals with no coronary plaque; as determined by CT angiography. A 1:1 nearest neighbor matching algorithm was employed to select 22 PLWoH controls, 22 PLWoH cases, 34 PLWH on ART controls, and 34 PLWH on ART cases (Fig. 5a). Cases and controls within each group were matched for age, ethnicity, BMI, CD4 T cell counts, and nadir CD4 T cell counts (Supplementary Table 7).

Adjusting for CVD risk factors (ACC/AHA Pooled Cohort Equation risk score) and aspirin use using logistic regression models, we found that levels of specific glycan structures associated with anti-inflammatory activities, including the terminal galactosylated glycans and other glycans (such as FA2[3]G1), were lower in PLWH cases compared to the other three groups (Fig. 5b), consistent with previous research in the general population[46,47]. Conversely, levels of glycans

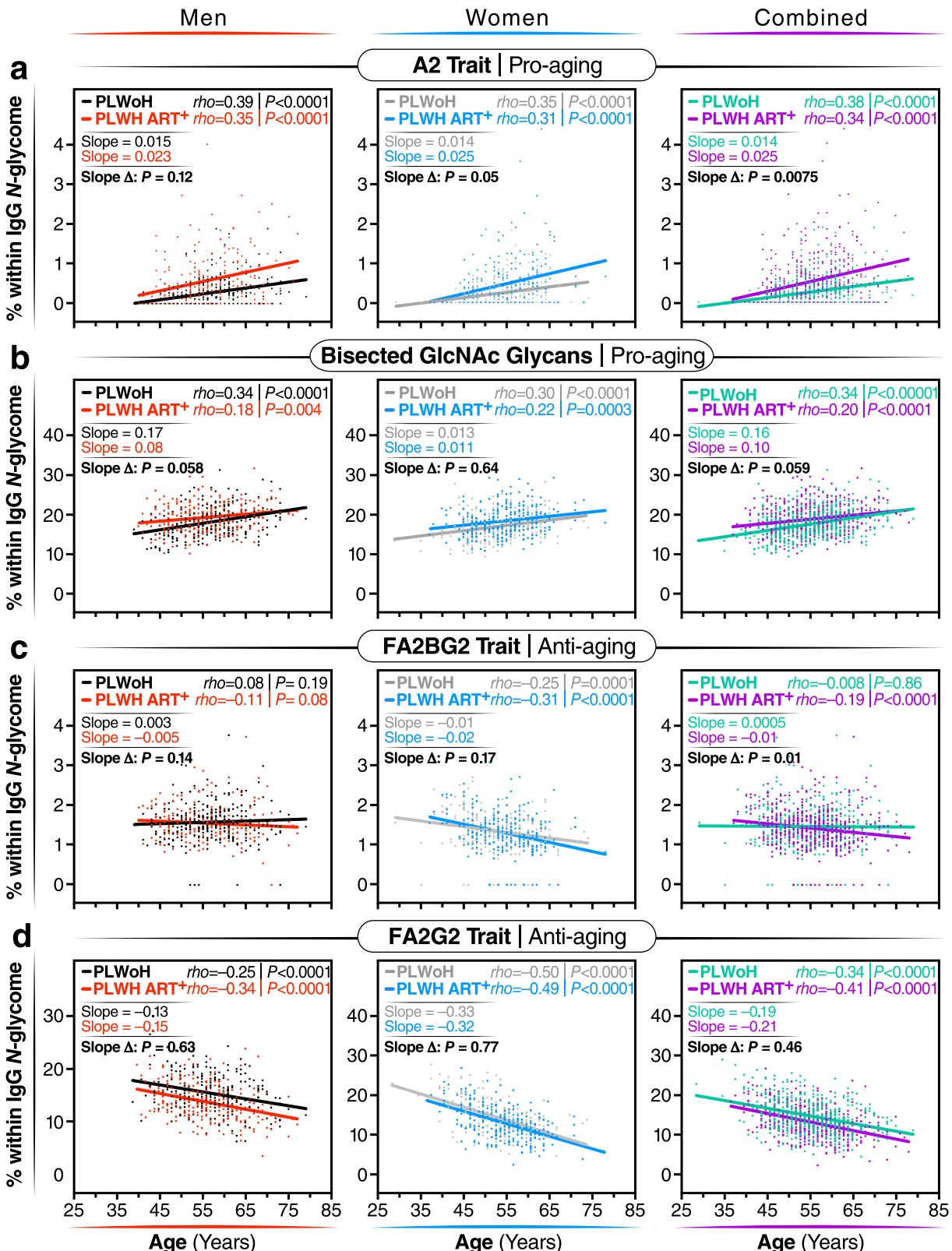

**Fig. 3 | The associations between IgG *N*-glycans and chronological age are altered in individuals with long-term ART-suppressed HIV infection.** Two-tailed Spearman's rank correlations between chronological age (x-axis) and the levels of the following glycan traits (y-axis): (**a**) A2 glycan trait, (**b**) bisected GlcNAc glycans, (**c**) FA2BG2 glycan trait, and (**d**) FA2G2 glycan trait, were analyzed separately for men, women, and combined groups. The significance of the difference between the slopes of the correlation in PLWH and PLWoH controls was determined from the interaction term between age and HIV status in a linear regression model. *N* = 985 biological samples. Source data are provided as a Source Data file.

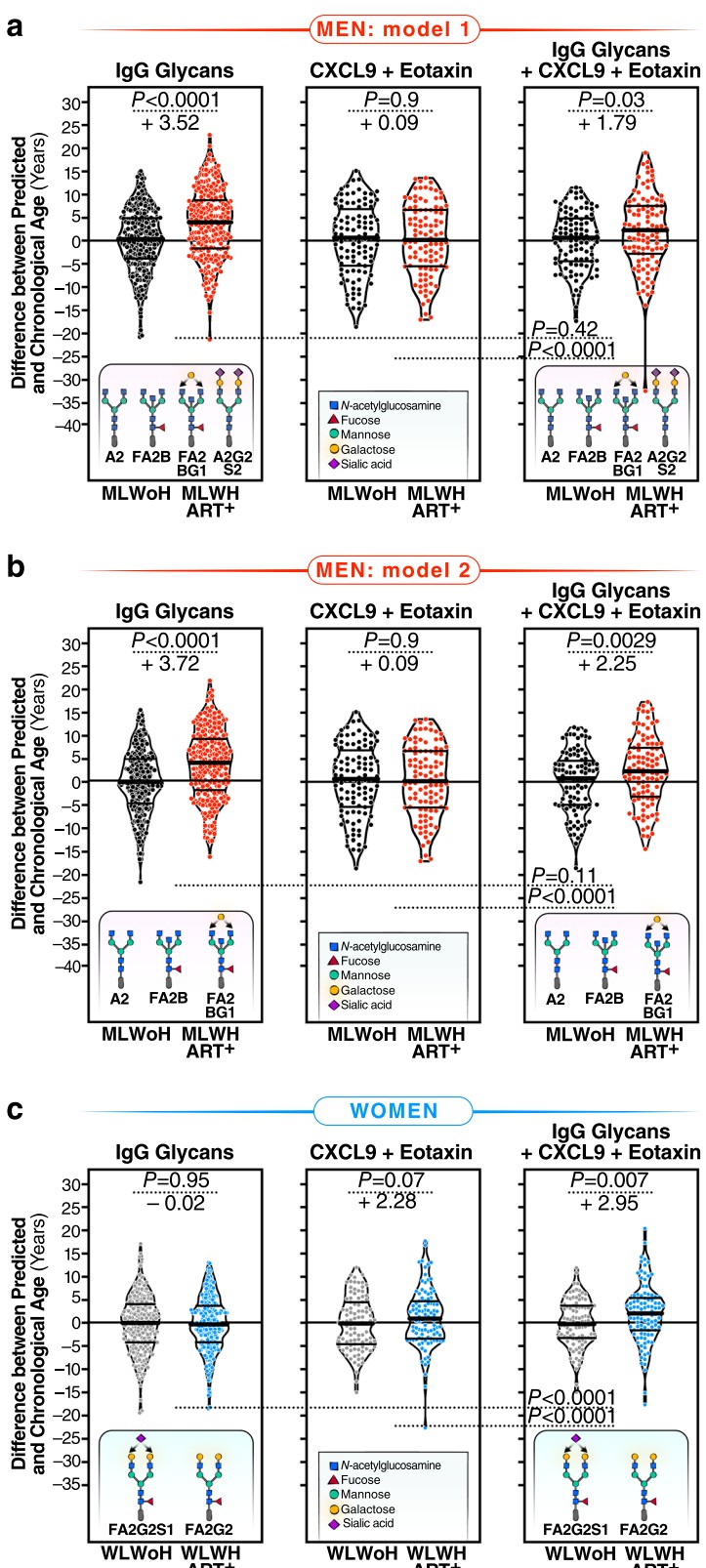

associated with pro-inflammatory activities, including agalactosylated glycans (the agalactosylated group and individual glycans lacking galactose such as A2, A2B, FA2, and FA2B) and the bisected GlcNAc group, were higher in PLWH cases compared to the other three groups (Fig. 5b). Furthermore, levels of many agalactosylated glycans were higher in PLWH with coronary plaque compared to those without coronary plaque, while levels of the individual galactosylated glycan

traits were lower in PLWH individuals with coronary plaque compared to those without coronary plaque (Supplementary Fig. 5).

We next examined the links between IgG glycans, plasma markers of inflammation (Supplementary Table 8), and the severity of CVD (measured by various scores depicted in Fig. 5c). We found that levels of IgG glycans associated with anti-inflammatory activities, mainly galactosylated glycans, were associated

**Fig. 4 | Machine-learning models based on IgG *N*-glycans indicate accelerated biological aging during HIV infection suppressed on ART.** Four (**a**) or three (**b**) specific glycan traits were selected using the LASSO model and included in models that correlated with chronological age in MLWoH, resulting in an average predicted age that closely matched chronological age. In MLWH, the model revealed an average acceleration of 3.52 (**a**) or 3.72 (**b**) years between predicted and chronological age when compared to their PLWoH counterparts. The left panels represent the model incorporating only *N*-glycans. The middle panels include two inflammatory markers, CXCL9 and Eotaxin. The right panels combine the *N*-glycan traits and the two inflammatory markers. Violin plots with median and interquartile range (IQR) are shown. *N* = 496 biological samples. **c** Two specific glycan traits were

selected using the LASSO model and included in a model that correlated with chronological age in WLWoH, resulting in an average predicted age that matched chronological age. The left panel represents the model incorporating only these two glycans. The middle panel includes the two inflammatory markers, CXCL9 and Eotaxin. The right panel combines the two glycan traits and the two inflammatory markers. In WLWH, only the model combining glycans and inflammatory markers revealed an average acceleration of 2.95 years between predicted and chronological age. Violin plots with median and interquartile range (IQR) are shown. *N* = 496 biological samples. For all panels, significance was determined using two-tailed *t* tests. The efficiency of the models was evaluated using the likelihood ratio test for nested models. Source data are provided as a Source Data file.

with reduced inflammation and lower CVD severity scores. Conversely, levels of glycans associated with pro-inflammatory activities, mainly agalactosylated and bisected GlcNAc glycans, were correlated with increased inflammation and higher CVD severity scores (Fig. 5c). These findings suggest that HIV/ART-associated alterations in IgG *N*-glycans profiles, particularly the loss of galactose and the gain of bisected GlcNAc, are associated with the severity of subclinical coronary artery disease during ART-suppressed HIV infection.

## IgG agalactosylation may precede the development of comorbidities in PLWH

Previous studies in the general population indicated that changes in IgG glycosylation patterns can precede the onset of age-related diseases[24,48]. Therefore, we conducted an exploratory longitudinal case-control study to investigate whether HIV/ART-associated *N*-glycan alterations may precede the development of age-related diseases in PLWH on ART. We used longitudinal plasma samples that had been collected from ten PLWH on ART before the onset of gastrointestinal (GI) cancers, which are non-AIDS-defining cancers associated with inflammation. Control samples were also obtained from 13 age-, sex-, and ethnicity-matched PLWH on ART without cancers at the same time points (Fig. 6a and Supplementary Table 9). Significant differences (FDR < 0.05) were observed between cases and controls in the average over time of IgG *N*-glycan groups and individual IgG glycans prior to cancer onset. Specifically, levels of galactosylated and sialylated IgG glycan Groups, as well as several galactosylated individual glycan traits, were lower in cases compared to controls (Fig. 6b, c, d). Conversely, levels of the agalactosylated group and individual glycan traits lacking galactose (such as A2 and FA2) were higher in cases compared to controls, on average, 5–10 years before cancer onset (Fig. 6b, c, e). Previous research in the general population had identified a specific IgG *N*-glycan ratio called the Gal-ratio, which reflects the distribution of IgG galactosylation and serves as a prognostic biomarker for cancer incidence[49–51]. In our study, we calculated the Gal-ratio by determining the relative intensities of agalactosylated versus mono-galactosylated and di-galactosylated IgG *N*-glycans using a previously described formula[49–51]. Remarkably, we found that the Gal-ratio predicted the occurrence of GI cancer in PLWH during ART treatment (Fig. 6b). This exploratory study suggests that dysregulations in IgG *N*-glycan patterns that are associated with premature aging and with pro-inflammatory responses are not only prevalent during ART-treated HIV infection but also may precede the onset of inflammation-associated diseases in PLWH.

## PLWH exhibit heightened expression of senescence-associated glycan-degrading enzymes compared to PLWoH

Our next objective was to investigate the potential upstream mechanisms responsible for the IgG *N*-glycan alterations associated with living with HIV and/or ART use. Several factors can influence the glycosylation of IgG, including the expression of glycosyltransferases in B cells. Notably, FUT8 catalyzes the transfer of core fucose, MGAT3

catalyzes the transfer of bisected GlcNAc, B4GALT1 catalyzes the transfer of galactose, and ST3GAL1 catalyzes the transfer of sialic acid (Fig. 7a). On the other hand, the removal of sialic acid and galactose can be attributed to increased systemic levels of glycosidases (NEU1-4 and GLB1, respectively), which can remove these components from IgGs after their production by B cells (Fig. 7a)[52,53]. We explored these two potential explanations. First, we examined the impact of ART-suppressed HIV infection on the expression of the four aforementioned glycosyltransferases in B cells. To achieve this, we performed single-cell CITE-seq analysis on PBMC from eight PLWH on ART and eight PLWoH. Following pre-processing and unsupervised clustering to identify immune subpopulations, the B cell cluster was regrouped into five sub-clusters (Supplementary Fig. 6). Upon closer examination, two of these sub-clusters did not express high levels of the B cell markers CD19 and CD20 (Supplementary Fig. 6) and were therefore excluded from further analysis. We observed modest decreases in the levels of ST3GAL1 and B4GALT1 in B cells from PLWH on ART compared to PLWoH controls, which is consistent with the reduction in levels of IgG sialylation and galactosylation observed in Fig. 1 (Fig. 7b, c). By contrast, we did not observe any changes in the levels of FUT8 or MGAT3 (Fig. 7d–e).

Next, we explored the second hypothesis that the removal of sialic acid and galactose could also be attributed to increased levels of glycosidases that remove these components from IgGs after their production by B cells. These glycosidases may not necessarily be produced by B cells, but their expression could be systemic. Therefore, we first analyzed gene expression data from various publicly available datasets that examined the transcriptomic profiles of blood and tissue cells from PLWH (viremic and suppressed by ART) and PLWoH. These analyses revealed that even after ART suppression, HIV infection was associated with elevated expression levels of several sialidases (enzymes that catalyze sialic acid removal), such as NEU1, NEU2, and NEU4, in peripheral CD4$^+$ T cells and cells from adipose tissues compared to controls (Fig. 7f–h). Additionally, ART-treated HIV infection was linked to increased expression levels of genes encoding β-galactosidase (GLB1, an enzyme that catalyzes galactose removal) in tissues compared to controls (Fig. 7i). The observation of elevated β-galactosidase mRNA levels in individuals receiving ART for HIV infection was intriguing, as β-galactosidase (β-Gal) is a well-established marker of senescence and aging[54–56]. Therefore, we sought further to validate the elevation of β-galactosidase during ART-treated HIV infection, and we measured β-galactosidase protein activity in the plasma of 17 PLWoH individuals and 24 PLWH on ART. The results shown in Fig. 7j indeed demonstrated elevated active β-galactosidase protein levels during ART-treated HIV infection compared to controls. Consistently, levels of active β-galactosidase protein correlated with higher levels of agalactosylated IgG glycans and lower levels of di-galactosylated IgG glycans, especially in PLWH (Fig. 7K). Together, these findings suggest that decreased levels of certain glycosyltransferases in B cells and increased levels of systemic glycosidases, namely sialidase and β-galactosidase, may contribute, at least partially, to some of the HIV/ART-associated *N*-glycan alterations.

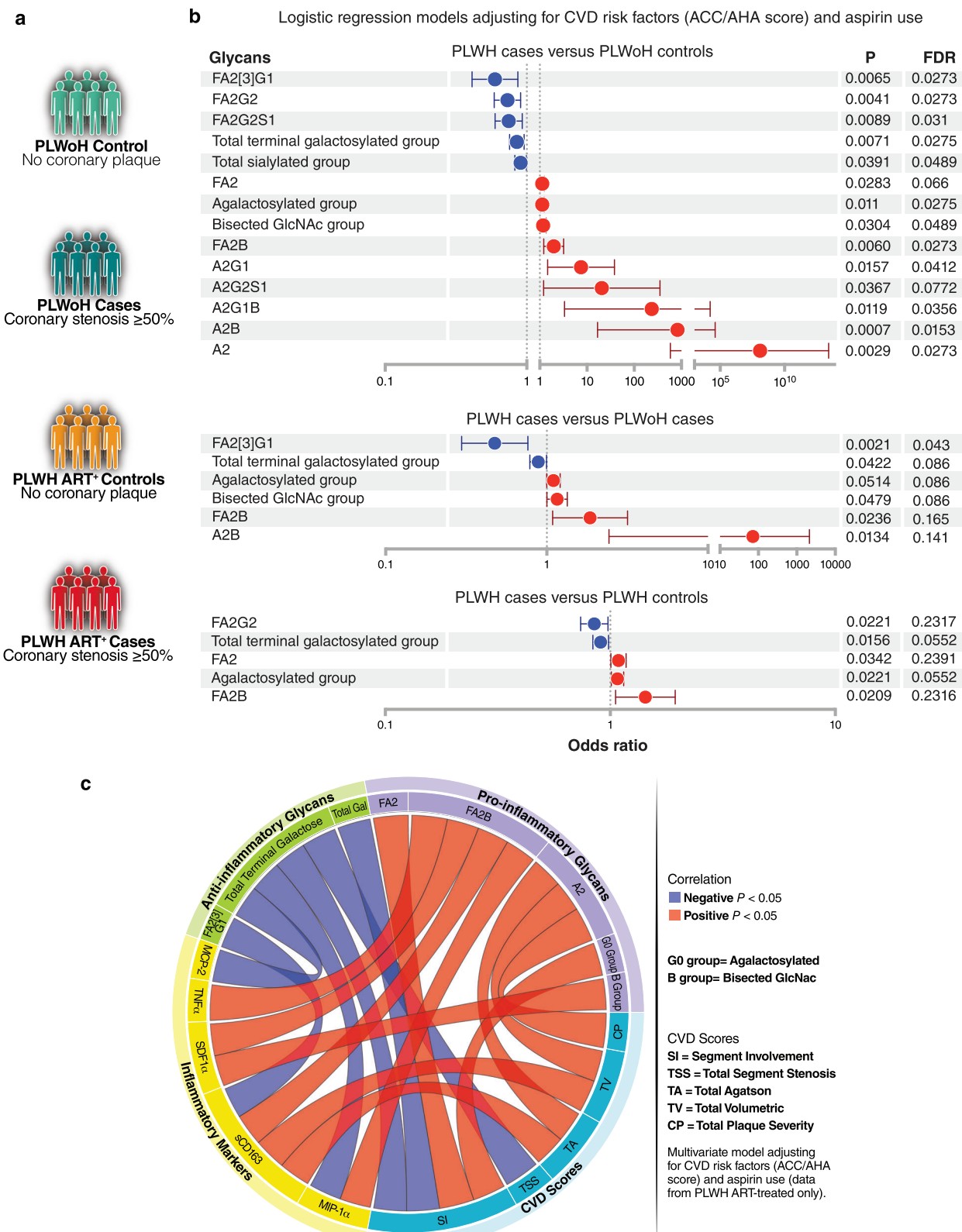

**c**

Correlation
- **Negative** *P* < 0.05 (blue)
- **Positive** *P* < 0.05 (orange)

**G0 group= Agalactosylated**
**B group= Bisected GlcNac**

**CVD Scores**
**SI** = Segment Involvement
**TSS** = Total Segment Stenosis
**TA** = Total Agatston
**TV** = Total Volumetric
**CP** = Total Plaque Severity

Multivariate model adjusting for CVD risk factors (ACC/AHA score) and aspirin use (data from PLWH ART-treated only).

## HIV/ART-associated IgG *N*-glycan alterations compromise HIV-specific, IgG Fc-mediated antiviral innate immune functions

Our final objective was to explore some of the potential downstream consequences of HIV/ART-associated IgG *N*-glycan alterations, which may mechanistically explain their association with worse clinical outcomes. We specifically aimed to assess whether these glycan alterations can influence IgG-mediated antiviral activity. In addition to viral

antigen neutralization, IgG, through its Fc domain, can engage various innate immune cells to trigger anti-viral innate immune functions. For instance, IgG can engage natural killer (NK) cells to induce ADCC, phagocytic cells to elicit ADCP, and complement proteins to elicit ADCD (Fig. 8a). The ability of an antibody to trigger these functions depends on several factors, including Fc glycosylation[10,57–59]. Extensive evidence has shown that the sugar fucose significantly reduces the

**Fig. 5 | IgG agalactosylation and bisecting GlcNAc correlate with the development and severity of coronary atherosclerosis during ART-treated HIV Infection. a** Schematic representation of the cases and controls included in this analysis. **b** Two-tailed logistic regression models illustrating the levels of glycans that exhibited significant differences between PLWH cases and any of the other three groups. The models were adjusted for CVD risk factors (ACC/AHA score) and aspirin use. $N = 22$ PLWoH controls; $N = 22$ PLWoH cases; $N = 34$ PLWH on ART controls; and $N = 34$ PLWH on ART cases. Dots represent the odds ratio, and the bars represent the 95% upper and lower limits of odds ratio. FDR values were calculated using the Benjamini-Hochberg procedure to correct for multiple comparisons. **c** A circos plot displaying two-tailed Spearman's rank correlations between glycans associated with anti-inflammatory activities (green), glycans associated with associated with pro-inflammatory activities (purple), plasma inflammatory markers (yellow), and CVD scores (teal). Red connections indicate a significant positive correlation, while blue connections indicate a significant negative correlation. Only correlations that remained significant after adjusting for CVD risk factors (ACC/AHA score) and aspirin use are shown. The correlation analysis includes only samples from PLWH on ART ($n = 68$). Source data are provided as a Source Data file.

ability of IgG to induce ADCC[60]. However, the role of galactose, or the lack thereof, in anti-HIV-specific innate immune functions remains unknown.

To investigate this, we selected samples from MLWH and WLWH and examined their ability to elicit anti-HIV specific ADCC, ADCP, and ADCD using in vitro functional assays. Initially, we compared the levels of ADCC and ADCP induced by equal amounts of bulk IgG (20 μg) from MLWH and WLWH, finding that bulk IgG from MLWH exhibited higher levels of ADCC and ADCP than did bulk IgG from WLWH (Fig. 8b). However, since these results could be influenced by the quantity of HIV-specific antibodies in this amount of bulk IgG, we measured levels of gp120-specific antibodies in these 20 μg samples and found that MLWH had higher levels of gp120-specific antibodies than WLWH (Fig. 8c). Therefore, we normalized the ADCC, ADCP, and ADCD abilities of bulk IgG based on the quantity of HIV-specific antibodies within these samples and found no differences between MLWH and WLWH in ADCC and ADCP. However, ADCD activity was lower in MLWH compared to WLWH (Fig. 8d).

Next, we examined the correlations between IgG glycosylation and the normalized ability of IgG to induce ADCC, ADCP, and ADCD. Considering that fucose plays a significant role in IgG-mediated innate immune functions, we focused on glycan structures that contained fucose, with and without galactose. As shown in Fig. 8e, f, fucosylated agalactosylated glycans exhibited negative correlations with ADCC. However, when the same fucosylated glycans were galactosylated, the correlations became positive, suggesting that galactosylation may improve these innate immune functions, and loss of galactose may compromise these functions. Indeed, total agalactosylated glycans negatively correlated with ADCC, while the galactosylated glycans positively correlated with ADCC. Additionally, the sialylated glycans correlated positively with ADCC and ADCP, and the bisected GlcNAc glycans correlated negatively with ADCP (Fig. 8e).

Beyond the correlations between antibody glycosylation and Fc-mediated innate immune functions, and to unequivocally demonstrate a mechanistic link between the loss of galactose and lower anti-HIV Fc-mediated innate immune functions, especially ADCC as shown in Fig. 8e, f, we used a chemoenzymatic method to engineer IgG glycans. We applied this method to the HIV broadly neutralizing antibody (bNAb) 10-1074, generating four distinct glycoforms (Fig. 9a, b). Since fucose is known to dramatically reduce ADCC[60], we designed these glycoforms so that two of them retain fucose, but one contains galactose (FA2G2), while the other lacks galactose (FA2). Meanwhile, the other two glycoforms lack fucose, but one contains galactose (A2G2), while the other lacks galactose (A2). This strategy allowed us to examine if galactose can alter ADCC, beyond the impact of fucose.

We first examined the purity of the four glycoforms using both capillary electrophoresis and mass spectrometry (glycomics and glycoproteomics). As shown in Supplementary Fig. 7, all four glycoforms are of high purity (>90%). In addition to their purity, these four glycoforms exhibit comparable binding to HIV antigens (Fig. 9c, d). However, they induce different innate functions: among the fucosylated glycoforms, the glycoform containing galactose (FA2G2) demonstrated stronger ADCC than the corresponding glycoform lacking galactose (FA2). Similarly, among the non-fucosylated glycoforms, the glycoform containing galactose (A2G2) demonstrated stronger ADCC than the corresponding glycoform lacking galactose (A2) (Fig. 9e, f). Consistently, the galactosylated glycoforms enable more efficient ADCC-mediated reduction of the levels of HIV p24 in the cultures compared to non-galactosylated glycoforms, within both fucosylated and non-fucosylated glycoforms (Fig. 9g). It is worth noting that the glycosylation and function of wild type 10-1074, currently used in HIV cure clinical trials, are similar to the weak FA2 glycoform (Supplementary Fig. 7). These data suggest that the loss of galactose reduces anti-HIV Fc-effector functions, which is consistent with our in vivo data showing that this loss of galactose is associated with worse clinical outcomes and higher inflammation in PLWH.

## Discussion

This study is a comprehensive and controlled investigation examining the relationships between bulk IgG $N$-glycans, chronic inflammation, biological aging, and the development of comorbidities in PLWH effectively managed by ART. Our findings suggest that chronic HIV infection, even when suppressed with ART, disrupts the normal progression of age-associated IgG $N$-glycan alterations. These disruptions are linked to increased inflammation, accelerated biological aging, and heightened susceptibility to age- and inflammation-associated comorbidities. These findings have the potential to pave the way to identify glycan-based biomarkers of inflammaging during HIV infection. Furthermore, they can offer valuable insights into the underlying mechanisms driving age- and inflammation-associated diseases in PLWH.

While living with HIV was associated with significant alterations in IgG $N$-glycans, we have observed a wide range of IgG $N$-glycan profiles among PLWH and their controls. IgG $N$-glycan profiles can be influenced by various demographic, environmental, and clinical factors, including genetics, diet, age, sex, gender, pregnancy, smoking status, chronic conditions (such as diabetes), and medications[11,16,30,61–63] For instance, we observed a significant influence of menopause status on the HIV/ART-associated IgG $N$-glycan alterations. Future investigations should examine the impact of each of the aforementioned factors, as well as CMV infection status, ART regimen, ART duration, and gender, on IgG $N$-glycans during ART-suppressed HIV infection. This may help identify specific clusters of individuals with distinct $N$-glycan profiles. In our exploratory analysis, we also found that IgG $N$-glycan alterations may precede the development of non-AIDS-defining cancers in PLWH on ART. However, this observation requires validation in larger studies with longitudinal samples from PLWH who have documented comorbidity outcomes. Such validation, along with the aforementioned clustering analysis, could establish the foundation for using these IgG $N$-glycan profiles individually or in combination (as in a model) as biomarkers to predict the development of aging- and inflammation-associated comorbidities. Such biomarkers may contribute significantly to improving the clinical management of individuals living with chronic viral infections.

In addition to serving as biomarkers for accelerated biological aging and the development of aging- and inflammation-associated diseases[16,64–68], IgG glycans are biologically active molecules playing significant roles in mediating immunological functions[12,69,70]. During

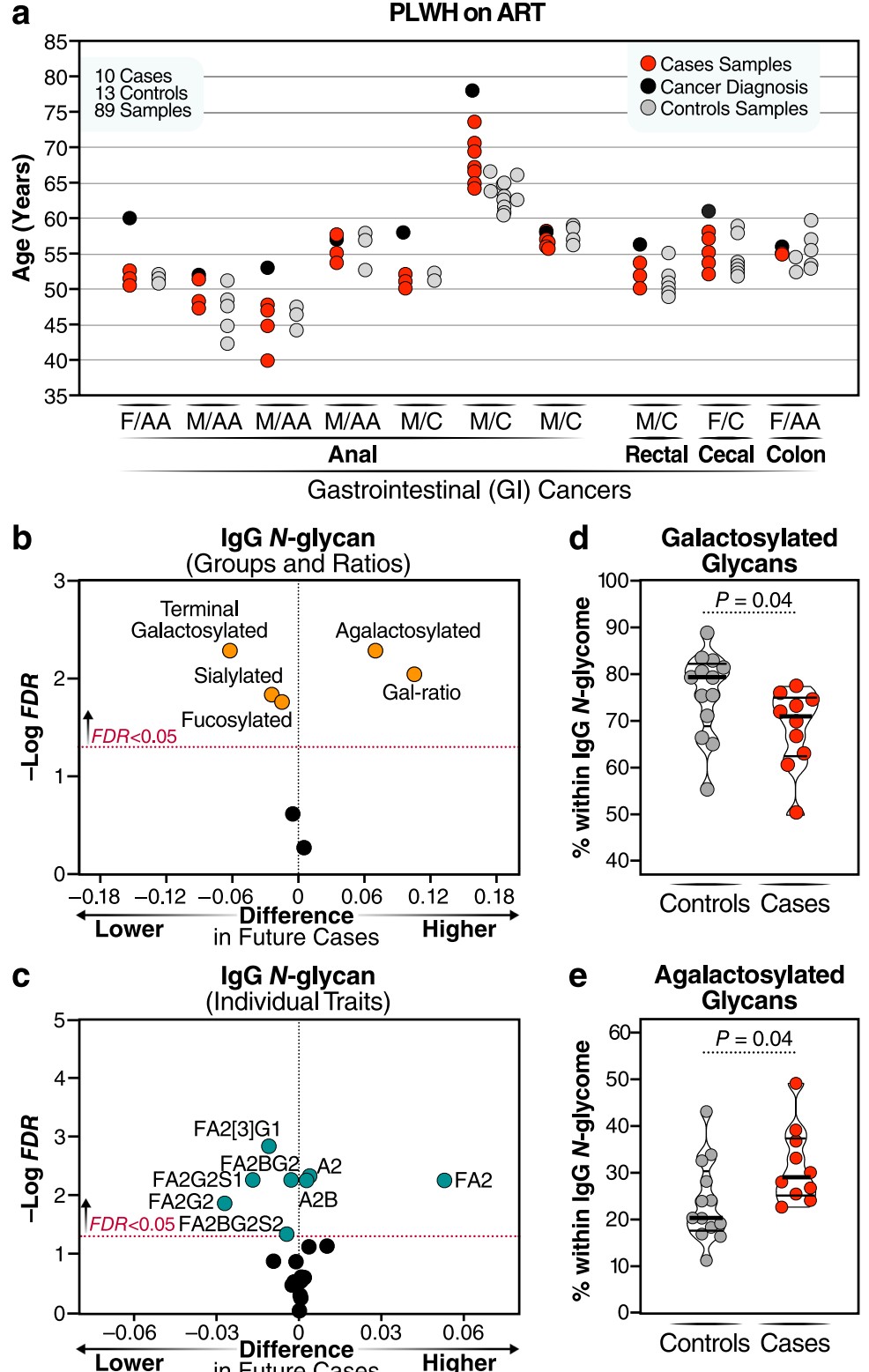

**Fig. 6 | IgG agalactosylation may precede the development of GI-related cancers in PLWH on ART. a** Overview of the longitudinal plasma samples analyzed from cases and controls, including the corresponding ages at the time of collection. The labels F and M represent female and male, respectively, while AA and C indicate African American and Caucasian ethnicity. Samples from cases and controls were matched based on age, sex, and ethnicity. Volcano plots illustrating the IgG glycan groups (**b**) and individual IgG N-glycans (**c**) that exhibited a significantly higher (right) or lower (left) average over time in cases compared to controls. FDR values were calculated using the Benjamini-Hochberg procedure to correct for multiple comparisons. Comparison of the average levels of total terminal galactosylated (**d**) and agalactosylated (**e**) IgG over a 5–10 year period before cancer onset in cases, contrasting cases with controls. Median and IQR are displayed. Two-tailed unpaired t tests were employed for statistical analysis. Source data are provided as a Source Data file.

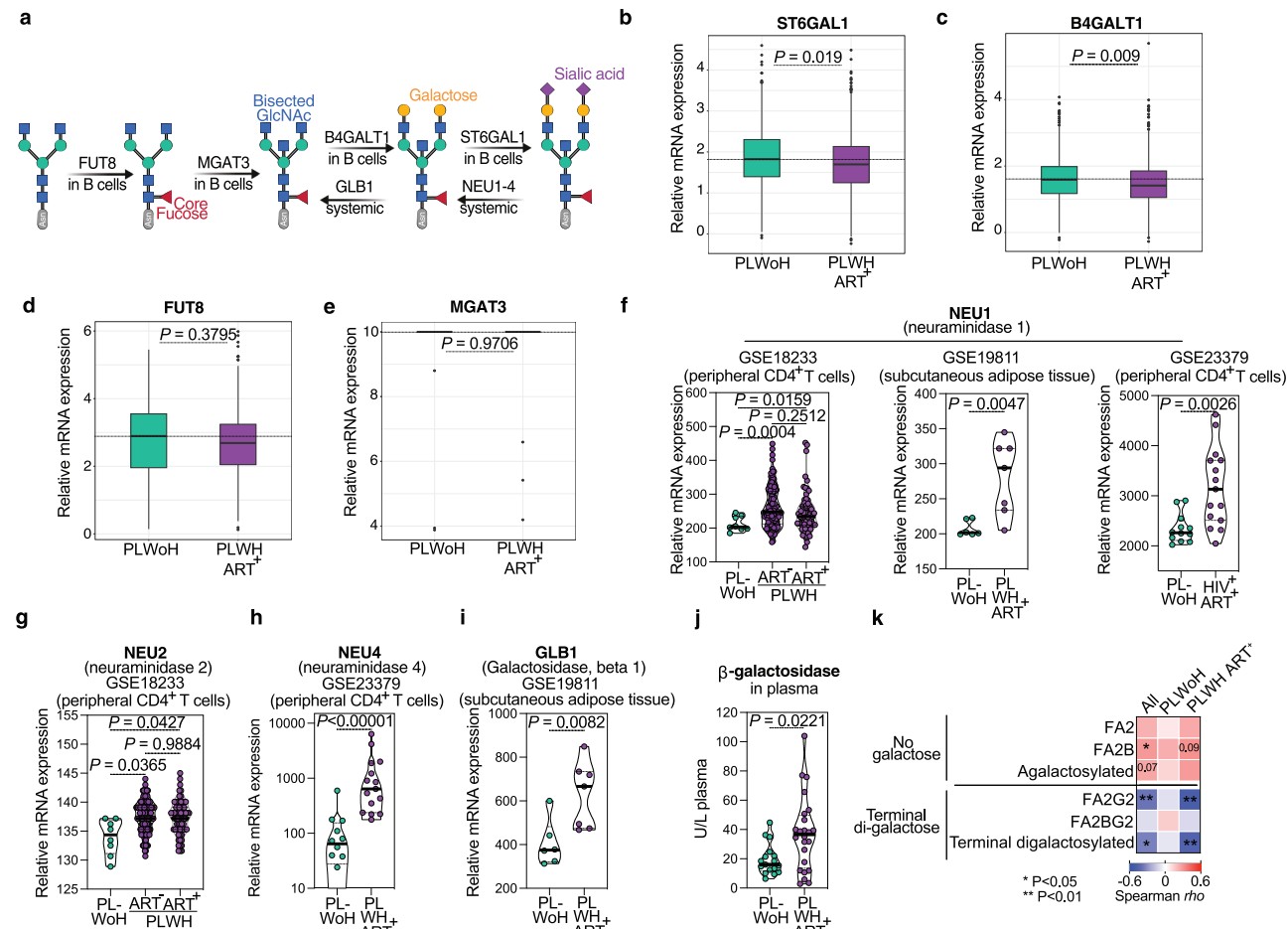

**Fig. 7 | PLWH exhibit heightened expression of senescence-associated glycan-degrading enzymes compared to PLWoH. a** Schematic representation illustrating the glycosyltransferases responsible for catalyzing different glycans into IgG $N$-glycans, as well as the glycosidases that remove certain glycans from IgG glycans. Single-cell expression levels of ST3GAL1 (**b**; data from 1050 single cells), B4GALT1 (**c**; data from 1295 single cells), FUT8 (**d**; data from 469 single cells), and MGAT3 (**e**; data from 33 single cells) in B cells from PLWH on ART and PLWoH. Single cells data were derived from samples of $n = 16$ PLWH on ART. Two-tailed unadjusted unpaired $t$ tests were performed for statistical analysis. Median and interquartile range (IQR; 25%–75%) are presented. Analysis of publicly available gene expression datasets (GSE18233, GSE19811, GSE23379) for genes encoding sialidases (NEU1 (**f**), NEU2 (**g**),

and NEU4 (**h**)) and β-galactosidases (GLB1 (**i**)), comparing their expression levels between PLWH and PLWoH. Unpaired $t$ tests were performed for statistical analysis. Two-tailed ANOVA or Mann–Whitney tests were performed for statistical analysis, and violin plots display the median and IQR. **j** Measurement of active β-galactosidase protein levels in the plasma of PLWH on ART and PLWoH. Two-tailed Mann–Whitney T-test was performed for statistical analysis, and violin plots display the median and IQR. **k** Two-tailed Spearman's rank correlation heatmap reveals the associations between plasma β-galactosidase in all donors, PLWoH, or PLWH on ART, and agalactosylated or galactosylated IgG glycans from the same donors. Positive correlations are depicted in red, while negative correlations are shown in blue. *$P < 0.05$, and **$P < 0.01$. Source data are provided as a Source Data file.

ART-suppressed HIV infection, notable alterations in IgG glycosylation include the loss of galactose (agalactosylation) and sialic acid (hyposialylation). These changes are associated with increased inflammation, a higher incidence and severity of subclinical atherosclerosis, and inflammation-associated cancers in PLWH on ART. Potential mechanistic links between IgG agalactosylation or hyposialylation and increased inflammation in PLWH are illustrated in Supplementary Fig. 8. As shown in Fig. 9, IgG galactosylation enhances Fc-mediated anti-viral activities of antibodies, including ADCC, ADCP, and ADCD. Conversely, IgG agalactosylation diminishes these critical anti-viral immune functions. This reduction in anti-HIV IgG Fc-effector functions might contribute to inadequate control of virally infected cells, particularly in tissues. The effectiveness of ART in completely suppressing viral replication, especially in tissues where ART penetration may be sub-optimal, remains unclear[71]. It can be hypothesized that the compromised anti-HIV innate immune function resulting from agalactosylation could contribute to higher HIV persistence and consequently greater inflammation. This hypothesis is further supported by our

previously observed negative correlations between the degree of bulk IgG galactosylation and the levels of cell-associated HIV DNA and RNA in CD4+ T cells during ART-suppressed HIV infection[72].

HIV-associated IgG agalactosylation can also directly lead to inflammation. IgG galactose facilitates interactions between FcγRIIB (CD32b) and dectin-1, triggering anti-inflammatory cascades[14,73,74]. Conversely, IgG agalactosylation is linked to pro-inflammatory functions by inhibiting these cascades[14,73,74]. Finally, sialic acid on IgGs (IgG sialylation) binds to receptors on myeloid cells, such as non-classical Fc receptors or sialic acid binding proteins (siglecs), initiating an inhibitory signal leading to anti-inflammatory responses through the inhibition of TLR4 signal transduction and consequent modulation of cytokine production[75–80]. The hypothesized effect of HIV-associated hyposialylation is to induce inflammation by reducing the potential for these anti-inflammatory cascades. Further experiments are needed to validate this model. These studies could inform the optimization of glycoengineering for broadly neutralizing antibodies in HIV cure and prevention strategies. Techniques like gene editing and metabolic

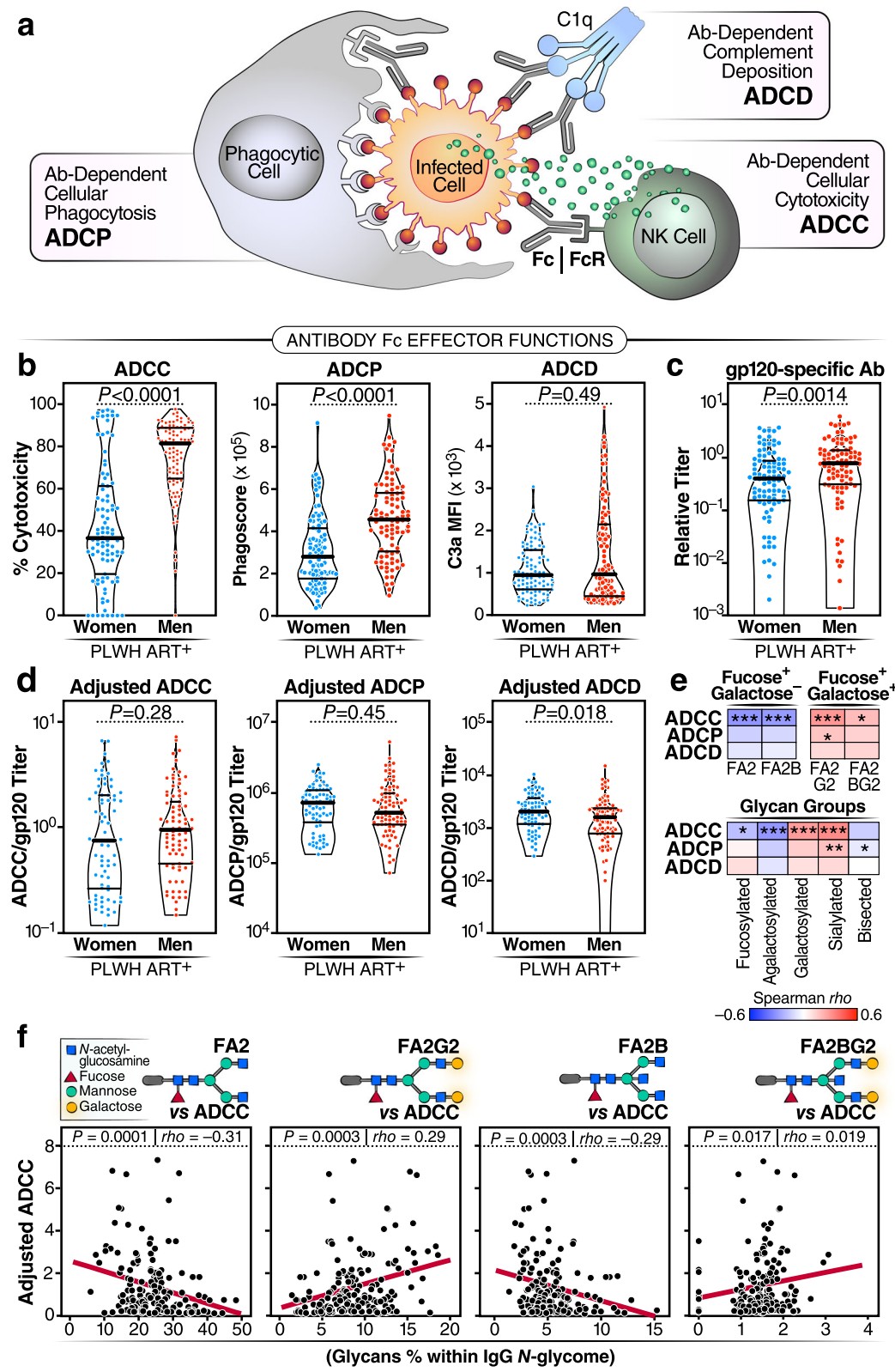

glycosylation inhibition have been used to modulate antibody interactions with Fc receptors and enhance ADCC activity. Understanding specific *N*-glycan traits impacting HIV or immune function could lead to glycan-based strategies to boost immune function during ART-suppressed HIV infection.

There are two potential explanations for the IgG *N*-glycan alterations associated with living with chronic HIV infection. The first

explanation is based on research suggesting that HIV infection leads to certain irreversible defects in B cells, even with ART. These defects may involve changes in the transcriptional profile of glycosyltransferases[81], which can result in differential antibody glycosylation. Notably, our previous studies showed that type-I interferons can induce elevated levels of bisected GlcNAc on bulk IgG during ART-suppressed HIV infection[82]. Cytokines, such as interferons, which are known to play a

**Fig. 8 | HIV/ART-associated IgG N-glycan alterations are linked to compromised HIV-specific IgG Fc-mediated antiviral innate immune functions. a** Schematic illustration highlighting the role of IgG Fc in facilitating various anti-viral innate immune functions, including ADCC, ADCP, and ADCD. **b** Evaluation of ADCC, ADCP, and ADCD elicited by 20 μg of bulk IgG from WLWH and MLWH after background subtraction using samples from WLWoH and MLWoH, respectively. **c** Measurement of gp120-specific antibody levels in the same amounts of bulk IgG from WLWH and MLWH. **d** Normalized data of ADCC, ADCP, and ADCD relative to the levels of gp120-specific antibodies measured in the same amounts of bulk IgG from WLWH and MLWH. For (**b**–**e**), violin plots display median and IQR, and

statistical significance was determined using two-tailed unpaired $t$ tests. $N = 157$–$187$ biological samples. **e** Two-tailed Spearman's rank correlation heatmaps demonstrate the associations between IgG glycans (columns) and adjusted ADCC, ADCP, and ADCD (rows). Positive correlations are represented in red, while negative correlations are depicted in blue. $*P < 0.05$, $**P < 0.01$, and $***P < 0.001$. **f** Examples from Two-tailed Spearman's rank correlations ($n = 161$ biological samples) in (**e**) illustrate that fucosylated glycans lacking galactose exhibit negative correlations with adjusted ADCC, while fucosylated glycans containing galactose display positive correlations with adjusted ADCC. Source data are provided as a Source Data file.

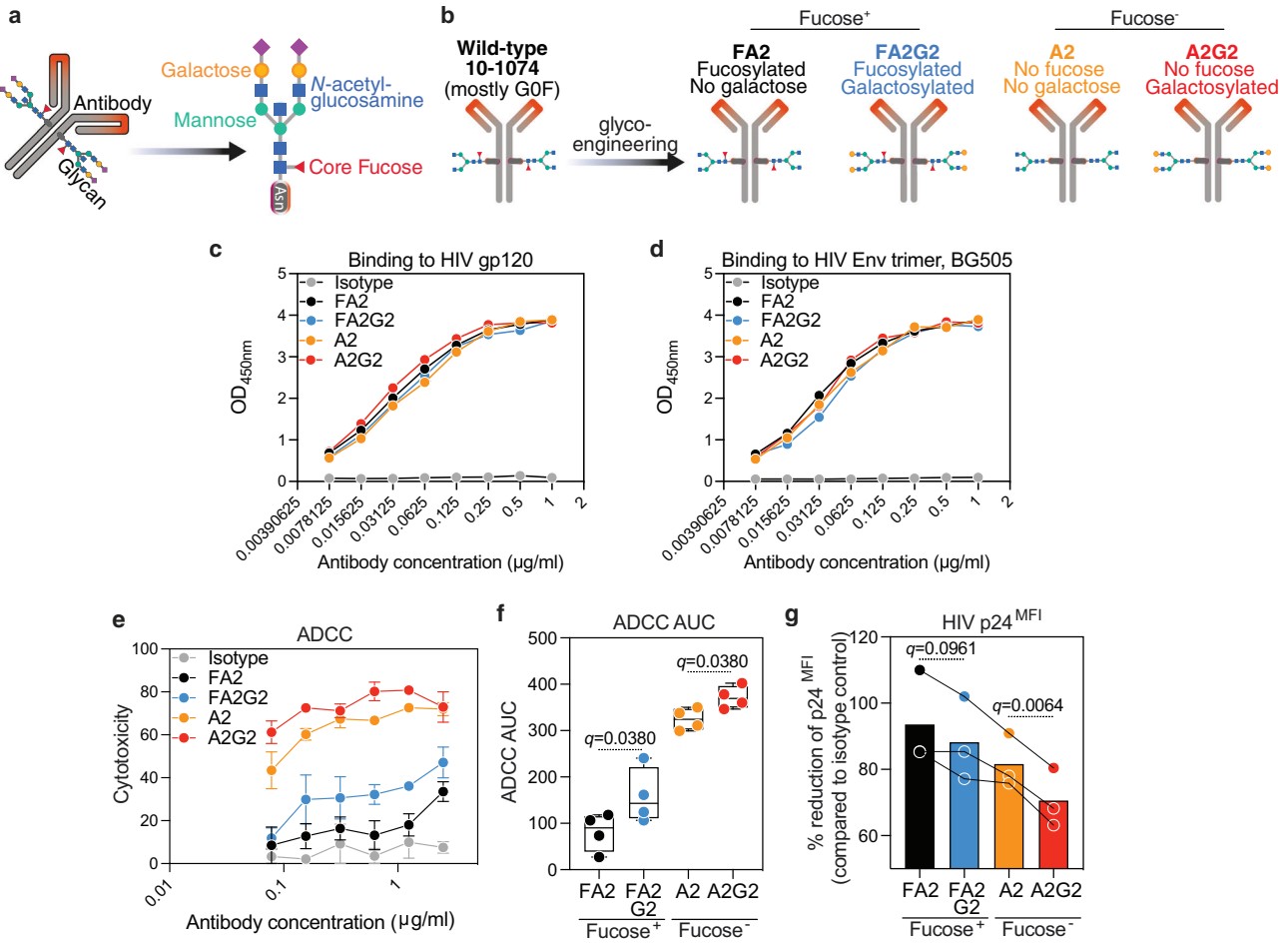

**Fig. 9 | Glycoengineering reveals that IgG agalactosylation reduces anti-HIV ADCC, while galactosylation enhances it. a** Schematic representation of IgG glycans. **b** Schematic illustration: 10-1074 was engineered by introducing glycans with four distinct glycan structures. The binding of the four glycoforms to HIV-1 gp120 (**c**) and HIV-1 Env trimer, BG505 (**d**), assessed by ELISA. Mean and standard error of the mean (SEM) are displayed. Among fucosylated glycoforms, the galactosylated glycoform (FA2G2) exhibited higher ADCC (**e**, **f**), resulting in lower levels of HIV p24 in infected cells (**g**) compared to the agalactosylated glycoform (FA2).

Among afucosylated glycoforms, the galactosylated glycoform (A2G2) exhibited higher ADCC (**e**, **f**), resulting in lower levels of HIV p24 in infected cells (**g**) compared to the agalactosylated glycoform (A2). $N = 3$–$4$ biological replicates. Error bars in (**e**) represents mean and standard SE. For (**f**), the box and whiskers showing all data from minimum to maximum. Analysis in (**e**, **f**) was done using one-way ANOVA corrected by the two-stage Benjamini, Krieger, & Yekutieli method. AUC = Area Under the Curve. Source data are provided as a Source Data file.

significant role in glycomic alterations associated with inflammatory diseases[83], could also be involved in modulating the glycosylation machinery of tissue plasma B cells, antibody glycosylation, and consequently, the functions of antibodies during ART-suppressed HIV infection. Further investigation is warranted to better understand the potential role of the cytokine milieu in this process during ART-suppressed HIV infection.

The second explanation for the changes in the IgG N-glycan patterns observed in HIV/ART-mediated conditions is based on research suggesting that there is an increased presence of glycosidases that

target sialic acid and galactose, and that these remove glycans from IgGs after the antibody is produced by B cells[52,53]. Our data support this hypothesis, as we observed higher expression levels of sialidase and β-galactosidase during ART-suppressed HIV infection. This observation that β-galactosidase mRNA levels are elevated in individuals receiving ART for HIV infection is intriguing for two reasons. Firstly, this increase in glycosidases may provide an explanation for the agalactosylation associated with HIV that we observed in our study. Secondly, β-galactosidase is a well-established marker of senescence and aging[54–56,84–86], In fact, both the genes encodes sialidase (NEU1) and β-

galactosidase (GLB) exhibited strong positive correlations with age in both mesenchymal and immune cells (FDR $p < 0.001$) in the general population, in a recent study[87]. This further reinforces the connection between HIV infection, accelerated aging, and the modulation of IgG N-glycan patterns. It is possible that increased levels of glycosidases, which lead to reduced levels of glycans associated with anti-inflammatory activities, such as sialic acid and galactose, could be a host response to the virus. That is to say, that the host may upregulate sialidase and β-galactosidases as a mechanism to decrease the anti-inflammatory effects of sialic acid and galactose, thereby promoting inflammation and enhancing the immune response to the infection. Additionally, this response of inducing glycan-degrading enzymes could be triggered by the chronic exposure to microbes from the gut microbiome, which occurs during HIV infection and is associated with translocation of microbes from the gut to the bloodstream. It has been demonstrated that myeloid cells express high levels of endogenous sialidases upon stimulation of Toll-like receptor 4 (TLR4) by bacterial lipopolysaccharide (LPS)[88]. These sialidases remove sialic acids from the surface of the myeloid cells, thereby reducing the potential for inhibitory interactions between Siglec receptors and sialic acids, which could dampen myeloid cell function[88,89]. While these mechanisms are likely important for maintaining a functional immune response to pathogens, they may also contribute to a state of inflammation. In support of this hypothesis, the loss of galactose from IgG N-glycans is not unique to HIV infection, and has been observed during other viral infections and conditions that involve microbial translocation, such as severe SARS-CoV-2 infection[15,90-95]. It is important to note that various Genome-Wide Association Studies (GWAS) showed a link between non-glycan related genes and IgG glycans[96], suggesting that the determinants of IgG glycosylation are likely multifactorial and involve the expression of several genes. Some of these genes might be directly related to the glycosylation machinery, while others may not.

Our study has limitations: (1) Since our study used samples from individuals with ART-suppressed HIV infection, we cannot examine the direct impact of HIV itself in the absence of ART nor other differences between groups that could confound the results. However, our previous studies demonstrated similar alterations in IgG N-glycans in HIV viremic individuals[72], suggesting that at least some of the alterations observed are mediated by the infection itself. Nonetheless, further research is required to determine if ART toxicity contributes to IgG glycan alterations, and this can be investigated, at least in part, using samples from Pre-Exposure Prophylaxis (PrEP) trials. Additionally, future studies should investigate whether early initiation of ART can prevent irreversible HIV-associated N-glycan alterations. (2) Our focus in this study was on women and men aged ~40 to over 65 years old. It is important to explore the impact of ART-suppressed HIV infection on younger and older individuals, particularly considering the potential influence of sex hormones in modulating IgG glycans. (3) It will be also necessary to investigate the effect of ART-treated HIV infection on N-glycan patterns in diverse populations, e.g., from various geographical regions, including low- and middle-income countries, as well as from areas with different HIV subtypes. (4) Our exploratory analysis in Fig. 6 suggests that IgG N-glycan alterations may precede the development of inflammation-associated diseases. However, it is essential to validate this possibility by conducting adequately powered longitudinal studies. (5) Detailed investigations into the causes and consequences of these IgG N-glycan alterations are necessary. These studies should encompass the analysis of the impact of HIV infection, ART, sex hormones, and inflammatory cytokines on the glycosylation machinery of tissue B cells and the activity of glycosidases in different immune cells. Our in vitro experiments with glycoengineered antibodies (depicted in Fig. 9) show that manipulating the levels of IgG glycans, which are modulated in PLWH, significantly alters the anti-viral Fc-mediated innate immune functions in the predicted directions−i.e., loss of galactose decreases anti-HIV ADCC, ADCP, and ADCD. However, more

investigation is essential to establish a definitive causal link between these glycan changes and the processes of immunity, inflammation, and aging in PLWH, beyond their potential utility as biomarkers. These studies should investigate the influence of glycan modulation on the Fc-mediated innate immune functions and inflammatory activities of antibodies in vivo, which can be accomplished through the use of animal models of HIV infection. (6) We have observed small changes in levels of IgG glycans between PLWH and controls; however, various studies have demonstrated that even subtle modulations in glycan structures can lead to significant variations in antibody activities and that minor glycan species can play a crucial role in antibody effector functions[12,81,97]. Despite these limitations, our current study represents a critical initial step towards elucidating the glycan-based mechanistic underpinnings of HIV-associated chronic inflammation in aging population.

## Methods
### Ethics, inclusion, and study cohorts
The study protocols were approved by the Institutional Review Board of The Wistar Institute (IRB protocol 21808309) as well as the Institutional Review Boards of Johns Hopkins University, University of Pennsylvania, University of Georgia, Albert Einstein College of Medicine, Emory University, SUNY Downstate Health Sciences University, University of Miami, University of Pittsburgh, University of North Carolina at Chapel Hill, University of California Los Angeles, Georgetown University Medical Center, University of California San Francisco, University of Alabama at Birmingham, Northwestern University, and Lundquist Institute of Biomedical Research at Harbor-UCLA Medical Center. Written informed consents were obtained from all participants. Study participants recruitment and/or sampling was not part of this study. All human experimentation was conducted per the guidelines set forth by the US Department of Health and Human Services and the authors' respective institutions. Detailed demographic and clinical information regarding the different study cohorts can be found in the results section, as well as in Table 1 and Supplementary Tables 4, 7, and 9. Menopause status was self-reported and classified based on the Stages of Reproductive Aging Workshop 10 years after the first workshop (STRAW + 10) criteria: (1) Premenopause (regular menstruation or menstruation with no persistent change that would meet subsequent criteria for peri- or postmenopause); (2) Perimenopause, including early perimenopause (defined as a persistent difference of seven days or more in the length of consecutive cycles) and late perimenopause (occurrence of amenorrhea of 60 days or longer but less than 12 months); and (3) Postmenopause (following 12 months of amenorrhea).

### IgG Isolation
IgG was purified from 50 μl of plasma using the Pierce Protein G Spin Plate (Thermo Fisher, catalog# 45204). Each well of the isolation plate contained 50 μl of resin, capable of purifying IgGs from up to 100 μl of serum. The protein G resin plate was discarded after each single use to avoid carryover and cross-contamination between samples. IgG was quantified using a BCA kit (Thermo Fisher, catalog# 23225) with bovine gamma globulin standard (Thermo Fisher, catalog# 23213). To confirm the purity of the IgG isolated using this method, we ran four random isolated IgG samples alongside a commercially available pure IgG sample from Sigma Aldrich (Catalog# I2511) on an SDS-PAGE and estimated the purity using densitometry analysis. High purity of the isolated IgG was observed, as shown in Supplementary Fig. 1c. For this analysis, IgGs were run under denaturing conditions using Bolt Bis-Tris Plus Mini Protein Gels, 4–12%, 1.0 mm (Invitrogen, catalog# NW04120BOX). The gel was run for 22 min at 220 volts. The gel was then washed three times for 5 min each with distilled water, followed by staining with Simple Blue (Invitrogen, catalog# LC6065) for 1 h. To further confirm the purity of isolated IgGs using this method, we used

Liquid Chromatography-Tandem Mass Spectrometry (LC-MS/MS) analysis on a random IgG sample along with the commercial IgG from Sigma Aldrich. As depicted in Supplementary Fig. 1d, this method confirmed the SDS-PAGE analysis, reaffirming the high purity of our IgG preparations and revealing a low-level contamination with other proteins. For this analysis, samples were electrophoresed into Bolt Bis-Tris Plus Mini Protein Gels, 4–12%, 1.0 mm (Invitrogen, catalog# NW04120BOX), and stained with Coomassie blue. The entire stained gel regions were excised, and in-gel digested with trypsin for LC-MS/MS analysis using a Q Exactive Plus mass spectrometer (Thermo Fisher Scientific) coupled to a Vanquish Neo UHPLC system (Thermo Fisher Scientific). MS data were searched against a UniProt human protein database (August 2023) and a common contaminant database using MaxQuant version 2.4.7.0[98]. Consensus identification lists were generated with false discovery rates set at 1% for protein and peptide identifications. Finally, to examine whether our IgG isolation method significantly impacted the percentage of different IgG subclasses, we analyzed the IgG subclasses in six paired plasma samples and isolated IgG from these samples using ELISA. As shown in Supplementary Fig. 1e, isolated IgGs contained the expected levels of IgG1 (approximately 60–70% of total IgGs), similar to the plasma, and about 20–30% IgG2-4, again similar to the plasma. For this analysis, levels of IgG1, IgG2, IgG3, and IgG4 were quantified using the IgG Subclass Human ELISA Kit (Invitrogen, Catalog# 99-1000) according to the manufacturer's instructions.

### IgG N-glycan analysis

IgG N-glycan analyses were performed using the GlycanAssure APTS Kit (Thermo Fisher, catalog# A33952), following the manufacturer's protocol. Specifically, IgG samples were incubated with the denaturing reagents for 5 min at 80 °C. After denaturing, samples were digested with PNGase enzyme for 10 min at 50 °C. Post-digestion, released glycans were labeled using APTS reagent and incubated for 60 min at 50 °C. The reaction cleanup was performed using positive selection with magnetic beads provided in the kit. To prepare the run, labeled and purified glycans were spiked in with GeneScan 600 LIZ size standard (Thermo Fisher; catalog# 4408399) to ensure consistent peak heights and to provide consistent results between different injections and capillaries. The N-glycans were analyzed using the 3500 Genetic Analyzer capillary electrophoresis system, as detailed in a previous publication[99]. Each sample was run in a single well, and samples were added to the machine in batches of 96 samples in a random order. Run parameters were: oven temperature at 60 °C, run time of 1330 s, injection time of 24 s, injection voltage of 1.6 kVolts, and run voltage of 19.5 kVolts. Identification of the glycan structures from relative migration units (RMU) was based on commercially available known glycan traits labeled with APTS evaluated using the same system by Thermo Fisher and others in several published and commercially available protocols[100-104]. The relative abundance of N-glycan structures was quantified by calculating the area under the curve of each glycan structure divided by the total area under the curve from all glycan peaks using the Applied Biosystems GlycanAssure Data Analysis Software Version 2.0. The identities of the peaks were based on already available libraries in the Applied Biosystems GlycanAssure Version 2.0 software. The software calculates a normalization factor based on a threshold setting. For each injection, the normalization factor is used as a multiplier to adjust the peak height of the sample peaks relative to the GS600 LIZ size standard peaks. The normalization factor has minimum and maximum limits, so if the size standard peak heights are abnormally high or low, the normalization will be limited. Peaks are then assigned to a glycan structure based on RMU, and the area under the curve is calculated automatically by the software. Several blank controls (composed of GeneScan 600 LIZ size standard, Landmark Red, and CE loading buffer) were run periodically. All peaks were assigned and identified, except for two peaks: a peak around RMU of

220, determined as a carryover based on blank samples (Supplementary Fig. 1a) and hence was not assigned; and another peak at around RMU of 334, which was assigned but not identified as different available libraries from Thermo Fisher and others have conflicted information on this peak's identity - some label it as A2G2, others as FA2BG1, and others do not assign it[100,101,104,105]. Therefore, to ensure rigor and reproducibility of our results, we elected to assign it but not identify it.

### Measurement of plasma inflammatory markers

Plasma levels of Fractalkine, IFN-α2a, IL-12p70, IL-2, IL-4, IL-5, IP-10, MCP-2, MIP-1α, SDF-1α, Eotaxin, IFN-β, IFN-γ, IL-10, IL-1β, IL-21, IL-6, leptin, CXCL9, and TNF-α were determined using U-PLEX kits from Meso Scale Diagnostics (Biomarker Group 1 (hu) Assays; catalog# K151AEM-2, Custom Immuno-Oncology Grp 1 (hu) Assays; catalog# K15067L-2, and Human MIG Antibody Set; catalog# F210I-3) according to the manufacturer's instructions. Soluble CD14 and soluble CD163 were measured by ELISA using DuoSet kits from R&D Systems (catalog# DY383 and DY1607, respectively) following the manufacturer's protocol.

### Measurement of β-galactosidase activity in the plasma

β-galactosidase (GLB1) activity in the plasma was quantified using the Beta-galactosidase Assay Kit (Abbexa; catalog# abx298865) following the manufacturer's protocol. Briefly, β-galactosidase in the plasma catalyzes the hydrolysis of nitrophenyl-β-galactopyranoside, producing nitrophenol. The concentration of nitrophenol is directly proportional to the enzyme activity in the samples, which can be determined by measuring the absorbance at 400 nm.

### Glycoengineering of 10-1074 antibody

The 10-1074 antibody was modified using several kits from Genovis. We utilized the TransGLYCIT G0 (Genovis catalog # T1-G0F-010) and TransGLYCIT G2 (Genovis catalog# T1-G2F-010) kits to create the glycoforms FA2 and FA2G2 from 10-1074. These kits selectively remove only the N-glycans on the Fc-part of the IgG, while preserving fucose. After deglycosylation, A2 glycan and A2G2 glycan were added to the core GlcNAc, followed by IgG purification. To generate glycoforms A2G2 and A2 without core fucose, we employed the TransGLYCIT G2 Afucosylated (Genovis catalog# T1-G2A-010) and TransGLYCIT G0 Afucosylated (Genovis catalog# T1-G0A-010) kits. These kits remove both the N-glycans on the inner GlcNAc and the α1−6 linked core fucose. A2 glycan and A2G2 glycan were then added, and IgG was purified in the subsequent step. To assess the purity of these modified glycoforms, we conducted capillary electrophoresis analysis as previously descried. We have also employed two mass spectrometry-based methods: (1) Glycoproteomics: IgGs were reduced with 25 mM dithiothreitol (DTT) at 60 °C for 40 min. The samples were further alkylated with 90 mM iodoacetamide (IAA) at room temperature for 20 mins. IgGs were desalted by 10 kDa centrifuge cartridges. Trypsin digestion was performed on reduced samples, 1:20/enzyme:protein ratio was used, and material was incubated at 37 °C for 18 h. Prior LC-MS/MS analysis, all peptide/glycopeptides were filtered using 0.2 μm filters[106]. The peptides/glycopeptides were analyzed on an Ultimate 3000 RSLCnano connected to a Thermo Eclipse mass spectrometer. Nano-LC columns of 15 cm length with 75 μm internal diameter, filled with 3 μm C18 reverse phase material were used for chromatographic separation. The separation conditions were low to high acetonitrile in a solution containing 0.1% formic acid, and the separation time was 60 min. The precursor ion scan was acquired at 120 k resolution in the Orbitrap analyzer and precursors at a time frame of 3 s were selected for subsequent fragmentation using stepped HCD (20, 30 and 40%). Charge state screening was enabled, and precursors with unknown charge state or a charge state of +1 were excluded. Dynamic exclusion was enabled (exclusion duration of 60 s). The centroided fragment ions were analyzed on an Orbitrap detector at 30 k resolution. The

resulting glycoproteomic data were processed in Byonic v5.2. and searched against the IgG1 sequence and a custom library of *N*-glycans from human plasma including 24 glycotypes. The precursor mass tolerance and fragment mass tolerances were set to 5 ppm and 10 ppm respectively. Additional modifications including deamidation of N and Q, carboxymethylation of C, and oxidation of precursor were also included in the search. Assignments were made using Byonic software and manual interpretation. IgG1 glycopeptide assignments was based on the following criteria: Log prob >1, mass error less than 2 ppm (in the most cases), and retention time. Relative abundancies were calculated from the raw data using m/z's corresponding to the different charge states. (2) Glycomics: 100 µg of IgGs were reduced with 25 µL of 25 mM DTT and incubated at 50 °C for 30 min. After that samples were treated with 90 mM iodoacetamide to alkylate. The samples were desalted by centrifugation via Amicon centrifuge filters (10 k MWCO size). 3 µL of PNGaseF was added to desalted samples followed by incubation at 37 °C for 18 h to release the *N*-glycans. The released glycans were permethylated using sodium iodide in the presence of sodium hydroxide and DMSO. Addition of water quenched the permethylation reaction and permethylated *N*-glycans were extracted with dichloromethane. The dichloromethane layer was rinsed five times with water and dried via evaporation by nitrogen gas[107]. Dried, permethylated glycans were re-dissolved in a solution of 100 µL of water and 100 µL of methanol for a total volume of 200 µL (with 1 mM NaOH). Samples were then run on a Themo Orbitrap Fusion Tribrid tandem MS coupled to Ultimate 3000 RSLCnano. 10 µL of *N*-glycans were injected into LC-MS/MS and ran in the low to high organic solvent gradient for 72 min. The precursor ion scan was acquired at 120 k resolution in the Orbitrap analyzer and precursors at a time frame of 3 s were selected for subsequent fragmentation using CID (45%). Charge states were targeted from 1 to 5. Dynamic exclusion was enabled (exclusion duration of 30 s). The centroided fragment ions were analyzed on an Orbitrap detector at 15 k resolution. GRITS software and Glyco Workbench 2 was used to identify the *N*-glycan structures (sodium adduct). Relative percentages were retrieved from MS1 level using m/z's for multiple charge states. FreeStyle 1.8 was used in order to pull out the peaks. Finally, to assess whether the modifications made to the 10-1074 antibody alter its binding to HIV, we performed examined their binding to HIV-1 gp120 and Env trimer BG505 using ELISA.

## Antibody-dependent cellular cytotoxicity (ADCC)

ADCC induction was estimated using CEM.NKR CCR5+ Luc+ cells (obtained through the NIH HIV Reagent Program, Division of AIDS, NIAID, NIH: CEM.NKR CCR5+Luc+ Cells, ARP-5198, contributed by Dr. John Moore and Dr. Catherine Spenlehauer. catalog# ARP-5198) co-cultured with primary human NK cells isolated from the blood of healthy donors using negative selection (Stem Cell; catalog#19055) in the presence of purified bulk IgG. HIV-infected and uninfected CEM.NKR CCR5+ Luc+ cells were washed and plated at $2 \times 10^4$ cells/well in a V-bottom microplate. The cells were treated with purified bulk IgG from study participants for 2 h at 37 °C. Control wells without antibodies were adjusted to volume with complete medium. After incubation, purified human NK cells were added to the wells, mixed, pelleted at 200 g for 2 min, and incubated for 16 h at 37 °C. To evaluate luciferase activity, 100 µl of supernatant was removed from all wells and replaced with 100 µl of Bright-Glo luciferase substrate reagent (Promega; catalog# E2610). After 2 min, the well contents were mixed and transferred to a clear-bottom black 96-well microplate. Luminescence (RLU) measurements were integrated over one second per well. The raw RLU values are shown relative to the light output generated in the medium-only control (background). For experiments involving glycoengineered glycoforms, following the ADCC assay, cells were fixed with 100 µL of Cytofix (Cytofix/Cytoperm Kit BD catalog # 554714) for 15 min at room temperature and permeabilized with 1X

Perm/wash buffer. The cells were then incubated for 30 min with the anti-HIV-1 core antigen (p24)-RD1 (Beckman Coulter catalog # 6604667, clone KC57), a gating example is shown in Supplementary Fig. 9a. After this incubation, the cells were washed and resuspended, and at least 100,000 events were acquired by flow cytometry using a BD Biosciences FACS Symphony.

## Antibody-dependent cellular phagocytosis (ADCP)

ADCP was measured using a flow cytometry-based phagocytosis assay[108]. Biotinylated recombinant HIV-1 gp120 from strain CN54 (Acro Biosystems; catalog# GP0-V182E6) was combined with fluorescent NeutrAvidin beads (Life Technologies; catalog# F8776) at a concentration of 1 µg protein per µl bead and incubated overnight at 4 °C. The beads were washed twice with 0.1% PBS-BSA to remove excess unconjugated antigens. The gp120-coated beads were resuspended in a final volume of 1 ml in 0.1% PBS-BSA. A 10 µl bead suspension was added to each well of a round-bottom 96-well culture plate, mixed with purified bulk IgG from study participants, and incubated for 2 h at 37 °C. Then, $5 \times 10^4$ THP-1 cells in 200 µl growth medium were added to each well and incubated overnight at 37 °C. The following day, 100 µl of supernatant from each well was removed, and 100 µl of BD Cytofix™ Fixation Buffer (BD; catalog# 554655) was added to each well. Cells were analyzed by flow cytometry, and the data collected were analyzed in FlowJo software. The percentage of fluorescent or bead-positive cells and the median fluorescence intensity of the phagocytic cells were computed to determine a phagoscore. A gating example is shown in Supplementary Fig. 9b.

## Antibody-dependent complement deposition (ADCD)

ADCD was assessed using a flow cytometry-based complement-fixing assay[109]. Briefly, $10 \times 10^6$ HUT78cells (obtained through the NIH HIV Reagent Program, Division of AIDS, NIAID, NIH: HUT 78 Cells, ARP-89, contributed by Dr. Adi Gazdar and Dr. Robert C. Gallo) were pulsed with 2 µg of recombinant HIV-1 gp120 from strain CN54 (Acro Biosystems; catalog# GP0-V182E6) for 20 min at 37 °C. Unbound gp120 was removed by washing the cells twice with 1% PBS-BSA buffer. Bulk IgG samples from study participants were added to the antigen-pulsed cells and incubated for another 30 min at 37 °C. Freshly resuspended lyophilized guinea pig complement (Cedarlane Labs; catalog# CL4051), diluted 1:20 with veronal buffer 0.1% gelatin with calcium and magnesium (Boston BioProducts; catalog# IBB-300), was added to the cells for 2 h at 37 °C. After washing with 1X PBS, the cells were assessed for complement deposition by staining with goat anti-guinea pig C3-FITC (MP Biomedicals; catalog# 0855385). After fixing, the cells were analyzed by flow cytometry, and ADCD was reported as the mean fluorescence intensity (MFI) of FITC-positive cells. A gating example is shown in Supplementary Fig. 9c.

## Measurement of gp120-specific antibodies

gp120-specific antibodies were measured by ELISA. Plates (Nunc maxisorp, flat bottom, Life Technologies; catalog# 44240421) were coated with 2.5 µg/mL of gp120 protein from strain CN54 (Acro Biosystems; catalog #GP4-V15227) overnight. The plate was washed with PBS-Tween20 and blocked with SuperBlock™ Blocking Buffer (Thermo Scientific; catalog #37515), and bulk IgG was added. Unbound IgG antibodies were washed, and bound IgG antibodies were detected using HRP-conjugated anti-human IgG antibody. After another wash, the plates were developed with TMB substrate (R&D; catalog #DY999) for 5–10 min in the dark, and the reaction was stopped using stop solution (R&D; catalog #DY994). The plates were immediately read at an optical density of 450 nm.

## Single-cell RNA-sequencing

Samples were uniquely barcoded using TotalSeq-B human hashtag antibodies (BioLegend), as per manufacturer's directions, to allow for

sample multiplexing for the 10x Genomics Chromium Controller single-cell platform (10x Genomics, Pleasanton, CA). Specifically, 2 million PBMC of each sample were first blocked with Human TruStain FcX (BioLegend; catalog# 422301), then incubated with 40 ng of various anti-human hashtag antibodies carrying unique cell barcodes and 12–100 ng of various CITE-seq antibodies, each previously optimized by flow cytometry. TotalSeq-B human CITEseq antibodies (BioLegend) used to subtype immune cells in the downstream data were: anti-human CD14 (catalog# 367145, clone 63D3), anti-human CD16 (catalog# 302063, clone 3G8), anti-human CD19 (catalog# 302263, clone HIB19), anti-human CD20 (catalog# 302361, clone 2H7), anti-human CD27 (catalog# 302851, clone O323), anti-human CD3 (catalog# 300477, clone UCHT1), anti-human CD38 (catalog# 356639, clone HB-7), anti-human CD4 (catalog# 300565, clone RPAT-4), anti-human CD56 (catalog# 362561, clone 5.1H11), anti-human HLA-DR (catalog# 307661, clone L243), anti-human CD45RA (catalog# 304161, clone HI100), and anti-human CD8 (catalog# 344757, clone SK1). After washing and dead cell exclusion using EasySep Dead Cell Removal (Annexin V) Kit (Stem Cell; catalog# 17899), a 10x G chip lane was super-loaded with a multiplexed pool of four uniquely barcoded samples, at a total of 40,000 single cells per lane with expected target recovery of 50%, five lanes were loaded in total. Single-cell droplets were generated using the Chromium Next GEM single cell 3' kit v3.1 (10x Genomics) on the 10x Genomics Chromium Controller. cDNA synthesis and amplification, library preparation, and indexing were done using the 10x Genomics Library Preparation kit (10x Genomics), according to the manufacturer's instructions. Overall library size was determined using the Agilent Bioanalyzer 2100 and the High Sensitivity DNA assay, and libraries were quantitated using KAPA real-time PCR. Each library consisting of four pooled samples was sequenced on the NextSeq 500 (Illumina) using a 75 base pair cycle sequencing kit (Illumina) and a paired-end run with the following run parameters: 28 base pair x 8 base pair (index) x 55 base pair.

Cell Ranger Suite (v3.1.0, https://support.10xgenomics.com) with refdata-cellranger-GRCh38-3.0.0 transcriptome as a reference was used to map reads on human genome (Hg38) using STAR[110]. Samples from eight PLWH on ART and eight PLWoH controls were combined and cells expressing fewer than 500 genes and genes that were not expressed in at least one cell were discarded. There were 61,114 cells left after initial quality control. Seurat v3[111] was used for clustering, marker identification, and visualization. Cell types were predicted against five human datasets as references using SingleR[112] and a consensus resolution of clusters into six immune cell types was made. Of these, the B cell cluster (5,186 cells) was subset and re-clustered. Gene expression was used to investigate if the sub-clusters were all indeed B cells. Two sub-clusters lacked in expression of known B cell markers and instead expressed genes associated with dendritic cells and T cells, likely due to spill over from neighboring clusters and/or ambient RNA. These were excluded from the analysis. The R package scCustomize was used to visualize gene expression of select markers as estimated kernel densities.

### Statistical analysis
The statistical methods used in data analysis for each figure are described in the corresponding figure legend. In general, two-group $t$ tests or Mann–Whitney tests were used to determine the difference between any two groups, and false discovery rates (FDRs) were calculated to account for multiple tests over the studied markers. Spearman's rank correlation analysis was used to evaluate correlations between variables. To determine the difference in the relationship between a studied marker and age, a linear regression model with main effects of age and HIV status, and the interaction term between age and HIV status, was applied. To evaluate the difference in studied markers among the combined conditions of HIV and CVD status, linear regression models were used with independent variables of HIV status,

CVD status, and their interaction term, adjusting for CVD risk factors (ACC/AHA score) and aspirin use. To explore which biomarkers of IgG $N$-glycans could be predictors of chronological age using data from PLWoH, who serve as representative counterparts to PLWH in the MACS/WIHS Combined Cohort Study, separately by sex assigned at birth, a linear regression model with the LASSO technique was first carried out using the 5-fold cross-validation (CV) selection option and the one-standard-error rule for determining the optimal tuning parameter. Due to the modest sample size, variable selection was determined using 100 independent rounds of CV LASSO. The biomarkers that were selected 80 or more times out of the 100 runs were used as the final set of predictors in our models.

### Reporting summary
Further information on research design is available in the Nature Portfolio Reporting Summary linked to this article.

### Data availability
The authors declare that data supporting the findings of this study are available within the paper and its supplementary information files. In addition, Mass spectrometry proteomic data were deposited to ProteomeXchange via the PRIDE database (project accession: PXD046510; DOI: 10.6019/PXD046510). Capillary electrophoresis data were uploaded to Zenodo (https://doi.org/10.5281/zenodo.10553464). Single cell RNAseq data were deposited to Gene Expression Omnibus (GEO) with accession # GSE254483. Access to individual-level clinical data from the MACS/WIHS Combined Cohort Study Data (MWCCS) is restricted due to HIPAA and MWCCS regulations regarding protection of sensitive data and may be obtained upon review and approval of a MWCCS concept sheet. Links and instructions for online concept sheet submission are on the study website. Source data are provided with this paper.

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

## Acknowledgements

This work is mainly supported by the NIH R01AG062383 and the NCI supplement to the Wistar Institute Cancer Center (P30 CA010815) to M.A-M. M.A-M. is also funded by the NIH grants, R01AI165079, R01NS117458, R01DK123733, R21AI170166, Penn Center for AIDS Research (P30 AI 045008), and the NIH-funded BEAT-HIV Martin Delaney Collaboratory to cure HIV-1 infection (1UM1AI126620) as well as the Gates foundation grant # Inv-033977. Mass spectrometry based glycomic analyses was partially supported by NIH R24GM137782 and GlycoMIP, a National Science Foundation Materials Innovation Platform funded through Cooperative Agreement DMR-1933525. We Would like to thank Drs. Michel Nussenzweig, Costin Tomescu, and Luis J. Montaner for providing the wild-type 10-1074 for the glycoengineering experiments and Dr. Daniel Kulp for providing HIV-1 Env trimer, BG505. Data in this manuscript were collected by the MACS/WIHS Combined Cohort Study (MWCCS). The contents of this publication are solely the responsibility of the authors and do not represent the official views of the National Institutes of Health (NIH). MWCCS (Principal Investigators): Atlanta CRS (Ighovwerha Ofotokun, Anandi Sheth, and Gina Wingood), U01-HL146241; Baltimore CRS (Todd Brown and Joseph Margolick), U01-HL146201; Bronx CRS (Kathryn Anastos, David Hanna, and Anjali Sharma), U01-HL146204; Brooklyn CRS (Deborah Gustafson and Tracey Wilson), U01-HL146202; Data Analysis and Coordination Center (Gypsyamber D'Souza, Stephen Gange and Elizabeth Topper), U01-HL146193; Chicago-Cook County CRS (Mardge Cohen and Audrey French), U01-HL146245; Chicago-Northwestern CRS (Steven Wolinsky, Frank Palella, and Valentina Stosor), U01-HL146240; Northern California CRS (Bradley Aouizerat, Jennifer Price, and Phyllis Tien), U01-HL146242; Los Angeles CRS (Roger Detels and Matthew Mimiaga), U01-HL146333; Metropolitan Washington CRS (Seble Kassaye and Daniel Merenstein), U01-HL146205; Miami CRS (Maria Alcaide, Margaret Fischl, and Deborah Jones), U01-HL146203; Pittsburgh CRS (Jeremy Martinson and Charles Rinaldo), U01-HL146208; UAB-MS CRS (Mirjam-Colette Kempf, Jodie Dionne-Odom, Deborah Konkle-Parker, and James B. Brock), U01-HL146192; UNC CRS (Adaora Adimora and Michelle Floris-Moore), U01-HL146194. The MWCCS is funded primarily by the National Heart, Lung, and Blood Institute (NHLBI), with additional co-funding from the Eunice Kennedy Shriver National Institute Of Child Health & Human Development (NICHD), National Institute On Aging (NIA), National Institute Of Dental & Craniofacial Research (NIDCR), National Institute Of Allergy And Infectious Diseases (NIAID), National Institute Of Neurological Disorders And Stroke (NINDS), National Institute Of Mental Health (NIMH), National Institute On Drug Abuse (NIDA), National Institute Of Nursing Research (NINR), National Cancer Institute (NCI), National Institute on Alcohol Abuse and Alcoholism (NIAAA), National Institute on Deafness and Other Communication Disorders (NIDCD), National Institute of Diabetes and Digestive and Kidney Diseases (NIDDK), National Institute on Minority Health and Health Disparities (NIMHD), and in coordination and alignment with the research priorities of the National Institutes of Health, Office of AIDS Research (OAR). MWCCS data collection is also supported by UL1-TR000004 (UCSF CTSA), UL1-TR003098 (JHU ICTR), UL1-TR001881 (UCLA CTSI), P30-AI-050409 (Atlanta CFAR), P30-AI-073961 (Miami CFAR), P30-AI-050410 (UNC CFAR), P30-AI-027767 (UAB CFAR), P30-MH-116867 (Miami CHARM), UL1-TR001409 (DC CTSA), KL2-TR001432 (DC CTSA), and TL1-TR001431 (DC CTSA). The MACS CVD2 study is funded by National Heart Lung and Blood Institute (NHLBI), R01 HL095129-01 (Wendy Post). The authors gratefully acknowledge the contributions of study participants and dedication of the staff at MWCCS sites.

## Author contributions

M.A-M conceived and designed the study. L.B.G. carried out the majority of experiments. O.S.A. and S.H.L. performed the ADCC, ADCP, and ADCD assays. S.S. and P.A. performed the mass spectrometry based glycomic analyses. T.K. and A.K. analyzed the single-cell cite-seq data. Q.L., X.Y., S.L., J.D., D.L., J.Z. and J.L.L.C.A. performed statistical analyses. D.B.H., I.O., J.L., M.A.F., S.H., B.M., A.A., B.D.J., C.R., D.M., N.R.R., O.K., S.G., S.W., M.W., W.S.P., A.L., I.F., P.C.T., R.G. and T.T.B. selected study participants and interpreted clinical data. L.B.G., Q.L., O.S.A. and M.A-M. wrote the manuscript, and all authors edited it.

## Competing interests

The authors declare no competing interests.

## Additional information

[1]The Wistar Institute, Philadelphia, PA, USA. [2]Cornell University, New York, NY, USA. [3]Johns Hopkins University, Baltimore, MD, USA. [4]University of Pennsylvania Perelman School of Medicine, Philadelphia, PA, USA. [5]University of Georgia, Athens, GA, USA. [6]Albert Einstein College of Medicine, Bronx, NY, USA. [7]Division of Infectious Diseases, Department of Medicine, Emory University School of Medicine, Atlanta, GA, USA. [8]SUNY Downstate Health Sciences University, New York, NY, USA. [9]Division of Infectious Disease, Department of Medicine, University of Miami, Miami, FL, USA. [10]University of Pittsburgh, Pittsburgh, PA, USA. [11]University of North Carolina, Chapel Hill, NC, USA. [12]University of California, Los Angeles, Los Angeles, CA, USA. [13]Georgetown University Medical Center, Washington, DC, USA. [14]Gladstone Institutes, San Francisco, CA, USA. [15]University of California San Francisco, San Francisco, CA, USA. [16]University of Alabama at Birmingham, Birmingham, AL, USA. [17]Northwestern University, Chicago, IL, USA. [18]Lundquist Institute of Biomedical Research at Harbor-UCLA Medical Center, Torrance, CA, USA. [19]Rush University, Chicago, IL, USA. ✉e-mail: mmohsen@Wistar.org

