## [Peer Review File · Nature Communications]

Editorial Note: Figure 1 in this Peer Review File has been amended to remove third-party material where no permission to publish could be obtained.

Reviewers' Comments:

Reviewer #1:

Remarks to the Author:

The manuscript by Giron et al. takes advantage of a well controlled human cohorts of PLWH and PNLWH to perform proteomic analysis. The main goals of the study is to profile the IgG glycome and associated with inflammation markers. The study is strong and scientifically robust and is well described. There are some limitations that are described below that should be remedied before publication. The major drawback of this study is that while differences are noted in all figures, the magnitude of these differences are not large. thus, it is unclear what the true biological meaning of these changes would be for PLWH.

The first major comment is that there are many differences in the immune system of PLWH that could contribute to a variety of comorbidities. However, these may be markers and not a true causative mechanism. The authors should better state this limitation and emphasize only the association of these traits with disease outcomes, including inflammation and aging. And limit the language that could imply causation.

The second major comment is that the authors do not explore the effects of ART alone on these parameters. Would individuals on PrEP have similar phenotypes. Are gene expression altered in these individuals for sugar enzymes? This should be explored or at the very least addressed. ART can impact the microbiome and virome and alter inflammation.

The third major comment is that while the authors point out and reference aberrant glycosylation has been shown in various disease states (mainly autoimmune), there is little evidence presented that IgG glycosylation is specifically associated with aging or immune age-related diseases. This should be better framed and demonstrated.

It makes sense that IgG glycosylation could change with age as a biomarker and that autoantibodies that are IgG targeting host proteins could be impacted by glycosylation, but how would someone with HIV (not an autoimmune disease) and have aberrant glycans on IgG affect the comorbidities the authors described. For example, how would the IgG glycosylation drive cardiovascular diseases in somebody without autoimmune disease. Moreover, how would the differential glycan on IgG impact age-related illness. Most IgGs are being directed towards pathogens so it could impede pathogen response but that is not the major statement by the authors.

Specific other points are below:

The authors classify IgG glycans as pro-aging and anti-aging. Could they provide data with the HIV-samples that show these levels correlate with age in their cohort. There are samples from a wide age range.

Figures and text should be appropriate PLWH and PNLWH.

For figure 1 panel E, the axis should go up to 100%.

Figure 2, leptin is not displayed but mentioned in the text.

Why are there no significant differences for soluble inflammatory markers for women compared with many for men?

Should there be two interaction P values one for women and men for figures 1 and 2. It appears some have an interaction that is changed by the other factors.

Figures 1 and 2 should be analyzed by sex as the authors did but also by age group as age is hypothesized to affect IgG glycosylation and the levels of inflammatory markers. Then the interaction could take into account BMI and other factors. This would eliminate the effect of age and sex on these markers and allow to see the true factors that are impacted by HIV infection.

It is unclear what the meaning or impact of 2J figure is. This should be better described in the text. It appears the data is very noisy.

Figure 5B-C needs to be revised. A hazard model should be applied and with hazard ratios performed. Also, only the ones that vary between the HIV+ Cases and HIV- cases should be emphasized as that is the main comparison. The remaining comparisons are useful but could be provided in supplementary material to avoid confusion. Panel C is very confusing and should be revised.

The magnitude of the differences in panes 6B-6D are concerning. It does not appear that the differences are large in magnitude. Also the p values should be displayed for panels D and E. Panels D

and E should also be tested for interaction from age or BMI or other factors that could impact this finding.

Panel 7B should be removed, it does not add to the analysis.

The levels of MGAT3 and FUT8 should also be displayed in 7C. With associated P values.

Why did the authors need to use public sequencing when they should also report the data from their CITE-seq. It is great if the public data confirms their data.

There most likely are datasets of PNLWH over the lifespan, do these mRNA levels change with age naturally.

Reviewer #2:

Remarks to the Author:

The manuscript "Plasma Glycomic Markers of Accelerated Biological Aging During Chronic HIV Infection" by Giron et al brings valuable new data about IgG glycosylation in HIV-positive individuals on anti-retroviral therapy. This study is worth publishing in Nature Communications assuming the following major and minor comments are properly addressed.

Gordan Lauc

Major comments

Ln232-241

LASSO model selected G0, G0FB, G1FB, and A2 glycans as the best predictor of chronological age. This aligns well with published data, but there is a caveat that should be checked. A2 glycan is a very minor component of the IgG glycome, while it is the major component of the transferrin glycome (PMID: 36959410). Since transferrin is the major contaminant of IgG during affinity capture, A2 glycan is often considered to be an indicator of contamination and it should not be assumed that it originates from IgG without additional confirmation that transferrin contamination was significantly

below 1% (which is not easy to achieve). I believe G2F glycan may be ranked just a bit lower than A2 in LASSO selection, and excluding A2 and adding G2F to the model may be an easy solution.

In Figure 3 Trait A1 is labelled as pro-ageing. Published data consistently show that sialylation decreases with age, thus this trait should be anti-ageing. Also, non-core fucosylated monosialylated glycans originate more from contaminating other proteins than from IgG, thus utmost care to estimate levels of contaminating proteins is needed before any claims about this structure can be made, in particular when the results go against previously published data.

From the description in Lns 274-277 it is not clear which glycans were analysed in this part of the study. The first sentence refers to “glycomic dysregulation linked with inflammation (Fig 3)”, but Fig 3 is showing age-related changes. Among the major IgG glycans the most extensive association with HIV+ART+ status was the extensive decrease in G1F[3] in women, which was recently identified to be the best predictor of future CVD events specifically in women (PMID 31915204 and 36174116). This glycan is not named here, but it is not clear whether it was not significant, or perhaps not analysed at all (i.e. grouped in G1F together with G1F[6]). Small number of women in this part of the study may also be the reason why it was not identified, but this should be clarified and perhaps also mentioned in the discussion.

Minor comments

Naming of glycans used in this paper follows commercial Luder nomenclature that is somewhat confusing in using A1 and A2 for mono and disialylated glycans. “Oxford notation” would be better to avoid confusion.

Section introducing functional aspects of IgG glycosylation would benefit from inclusion of recent data reviewed in PMID 37414906

LN 501-502

Authors implicate that the change in expression of glycosyltransferases may be the causative element controlling IgG glycosylation. However, a number of studies demonstrated limited correlation between glycosyltransferase expression and glycome composition. Furthermore, GWAS studies indicated that glycosyltransferases are only a small part of the regulatory network that control glycosylation (PMID 32128391).

Reviewer #3:

Remarks to the Author:

In this study, the authors seek to profile the N-glycans of plasma IgG in a large group of HIV patients undergoing anti-viral treatment and in a matching HIV negative control cohort not receiving treatment. The focus is to study the accelerated aging experienced by HIV patients receiving treatment; the rationale to focus on IgG N-glycosylation as a proxy for aging is that others have previously linked the IgG N-glycans in plasma to biological aging. Profiling and quantitation of the released and labelled N-glycans were performed using high throughput CE from which several comparative analyses of the IgG glycans were performed after grouping the individual N-glycans for terminal traits (galactose, agalactose, fucose, bisecting GlcNAc termini etc). Several other types of experiments were performed to follow up on the initial results from the IgG glycoprofiling including measurements of inflammatory markers, glycosylation enzymes and cytotoxicity on both the patient IgGs and also glycoengineered IgG species. Given the importance of the glycan profiling data of IgG from the donor groups (Figure 1-8, Table 2-3) and from the glycoengineered IgG (Figure 9) I have selectively focused on providing comments to this part which fits with my expertise profile.

Major comments:

While the IgG glycan data is well compared and presented across the many figures and tables it is difficult to assess the robustness of the actual data since the generated data are with the exception of Supplementary Figure S1 presented in an already annotated/interpreted (Fig 1B-F) or semi-interpreted form (e.g. Table 2, Supplementary Figure 7 etc).

1) Data transparency and data sharing: Given the critical reliance on the accuracy of the IgG glycan identification and quantitation for the conclusions of this work, all data (raw, processed and annotated/interpreted) must be made available to the community via publicly accessible repositories to allow for scrutiny and further interrogation by others as per good data sharing practices and conventions in the field. This applies to all CE and MS glycomics and glycopeptide data (and presumably also to the single cell transcriptomics data albeit not covered in my review comments). The text in the current Data Availability section is not appropriate as it does not allow for the expected data transparency and sharing.

2) Glycan identification and profiling: The equally critical method section of the IgG N-glycan analysis is weak and does neither allow the reader to fully understand each experimental step of the glycan analysis let alone reproduce the experiments, which are minimum expectations given the

importance of the IgG glycan profiling for the conclusions of the study. As an analyst, it is still unclear to me how a confident identification of each of the N-glycans was performed in this study; were retention time standards applied and how were they used to produce confident glycan structure identification down to the linkage/branching level? Were RT libraries generated or already available? Looking at SFig S1, one peak is labelled B334.99; this peak is also included in Table 2, but labelled as an undefined trait. Why is this undefined peak included in the glycan profile? Another major peak at 220 migration units is consistently abundant in all traces, but was neither assigned nor included in the N-glycan profile. Without analytical evidence and a detailed explanation of the analytical approach this reviewer is not convinced about the identities of the assigned peaks.

3) Glycan quantitation and cohort acquisition: An important aspect of this study is that 1200+ samples were quantitatively profiled for IgG glycosylation which is not a trivial analytical feat; if not performed appropriately, such large scale data generation could lead to systemic bias in the observed IgG glycan levels across sample sets. The manuscript does not discuss any analytical aspects or issues relating to this point. It should be detailed exactly how the sample cohort was acquired (e.g. were samples scrambled in their injection order and run in one single batch or were individual patient groups analyzed one by one across multiple batches?). Were each of the samples analyzed once or in multiple technical replicates? Were standards run between samples to calibrate for retention time. What were the CE settings? Were blanks run to rule out carry over? Were the baselines and AUCs manually determined or performed by a software? These details must be detailed in the method section and the discussion should discuss any potential analytical weaknesses and unintended biased that could be introduced by the experimental design and any efforts to mitigate such unwanted effects.

4) Sample preparation: The IgG N-glycan profiling relies blindly on the ability of a Protein G column to quantitatively and consistently capture all IgG glycoforms (and only IgG) before PNGase F digestion and profiling. 50 ul of plasma (in excess of 1 mg IgG) was used for the capture which is a high amount and may be pushing towards the capacity limit of the column if not careful? Can it be ruled out that some of the differences in IgG glycosylation observed across patient groups is in fact introduced at the protein G column stage such as i) co-purification of other glycoproteins, ii) IgG subtype switching i.e. IgG1 towards IgG2-4 or iii) a selective capture of some IgG glycoforms over others? The authors should demonstrate that the protein G column captures only and repeatedly IgG under the applied conditions since this is the only way to ensure the glycan data is in fact originating from plasma IgG. The corresponding method section should be expanded to detail the protein G isolation procedure (e.g. was the same protein G column used for multiple samples and, if so, how was carryover avoided?) and the deglycosylation and labelling reactions should be explicitly explained. Deglycosylation was presumably not performed of natively folded IgG as indicated in the current method section and how was the excess labelling reagent and the deglycosylated protein removed from the labelled glycans before CE analysis?

Minor: Title and Short Title, P1: Instead of "plasma glycomic markers" the title should for accuracy be changed to "IgG glycan markers". The manuscript repeatedly uses "glycomics" and "IgG glycomics/glycome". The -omics/-ome suffix should by convention be reserved for system-wide

analyses which is not performed in this study (IgG centric). Use instead IgG N-glycans or IgG N-glycosylation.

Abstract, P1: The manuscript is difficult to read due to the many abbreviations, consider if some abbreviations are unnecessary such as PWH (seems redundant given that MWH and WWH are also in use).

Introduction: Several references are quite dated (up to 10-15 years old, e.g. some amongst these references Ref6-9, 10-11, 12-14, 15-17, 26-27. Can more recent literature be cited?

Results:

L110 and L116 (and in Table 1): Are these cell counts measured in blood?

L117: Should be 20 individual N-glycan structures since B334.99 is not a glycan until proven otherwise (see comment above).

L157: Figure 1G: Given the relatively subtle changes observed in the IgG glycosylation in HIV patients the indicated glycan transformation is hugely exaggerated which should be pointed out to uninformed readers.

L159 and L187: What is meant by “higher markers” and “higher inflammation”? Systemic markers/inflammation better?

L169: leptin with lowercase.

L247: “no inflammatory markers alone correlated...”?

L285 and elsewhere: anti- and proinflammatory glycans. Better with glycans associated with anti-or proinflammatory conditions to avoid implying causative relationships?

L336: “...observed HIV/ART-mediated glycomic alterations” change to “...IgG N-glycan alterations associated with HIV/ART” to avoid implying direct/causative relationship? Same for L449.

L346: Unbiased to unsupervised?

Methods:

L625: Why was both Byonic and pGlyco used? Only Byonic was detailed below.

L633: 100ug of the material. What is meant by material? Purified IgG?

Figure 1: Do the number of individuals in Figure 1 add up to match 1,216 individuals mentioned in the abstract?

All figures: The core fucose in all depicted glycan structures should by convention be changed to facing the right-hand side of the N-glycan since this is an alpha1-6 linked (not alpha1-3) fucose.

Supplementary Figure S1: The y axis looks like absolute (not relative) FUs, please check.

We are very grateful to the reviewers for providing us with very helpful, consistent, and constructive feedback on our work, and for giving us the opportunity to revise and improve our manuscript, now entitled "*Immunoglobulin G N-glycan Markers of Accelerated Biological Aging During Chronic HIV Infection.*"

We have conducted new experiments, performed new analyses, and made several modifications throughout the manuscript and its figures to address all the reviewers' concerns. We have included detailed, point-by-point responses to each of these concerns, describing the corresponding changes in our manuscript. Our responses are highlighted in blue text to facilitate the review process.

Reviewer #1

Reviewer general comment: The manuscript by Giron et al. takes advantage of a well controlled human cohorts of PLWH and PNLWH to perform proteomic analysis. The main goal of the study is to profile the IgG glycome and associated with inflammation markers. The study is strong and scientifically robust and is well described. There are some limitations that are described below that should be remedied before publication. The major drawback of this study is that while differences are noted in all figures, the magnitude of these differences are not large. thus, it is unclear what the true biological meaning of these changes would be for PLWH.

Authors response: We appreciate the reviewer's positive evaluation of our study, recognizing it as scientifically robust and well-described. In response to the feedback, we have made several modifications throughout the manuscript, conducted new analyses, and added a new figure, as detailed below.

Regarding the magnitude of the differences observed, various studies have demonstrated that even subtle modulations in IgG N-glycan structures can lead to significant variations in antibody activities and that minor glycan species can play a crucial role in antibody effector functions. Below are examples of such studies:

1. Ackerman ME, Crispin M, Yu X, Baruah K, Boesch AW, Harvey DJ, et al. Natural variation in Fc glycosylation of HIV-specific antibodies impacts antiviral activity. *J Clin Invest.* 2013;123(5):2183-92. Epub 2013/04/09. doi: 10.1172/JCI65708. PubMed PMID: 23563315; PubMed Central PMCID: PMC3637034.
2. Kaneko Y, Nimmerjahn F, Ravetch JV. Anti-inflammatory activity of immunoglobulin G resulting from Fc sialylation. *Science.* 2006;313(5787):670-3. Epub 2006/08/05. doi: 10.1126/science.1129594. PubMed PMID: 16888140.
3. Bye, A. P. et al. Aberrant glycosylation of anti-SARS-CoV-2 spike IgG is a prothrombotic stimulus for platelets. *Blood* 138, 1481-1489, doi:10.1182/blood.2021011871 (2021).

Nevertheless, we have added this observation and the need for future studies to examine the impact of these changes on inflammation, immunity, and aging in people living with HIV in the Discussion section.

Major comments:

1. *The first major comment is that there are many differences in the immune system of PLWH that could contribute to a variety of comorbidities. However, these may be markers and not a true causative mechanism. The authors should better state this limitation and emphasize only the association of these traits with disease outcomes, including inflammation and aging. And limit the language that could imply causation.*

Authors response: We agree with the reviewer. While our manuscript presents data in **Figure 9** indicating that the main glycomic alterations observed in PLWH — specifically, the loss of galactose or agalactosylation — can mechanistically hinder the efficacy of anti-HIV antibodies in eliciting anti-HIV-specific Fc-mediated innate immune functions (ADCC, ADCP, and ADCD), we have elaborated on the need for further comprehensive investigations to demonstrate causal relationships between these glycomic alterations and both inflammaging and immunity in PLWH.

Specifically, in the limitations section of the Discussion, we have now added: "*Detailed investigations into the causes and consequences of these IgG glycomic alterations are necessary. These studies*

should encompass the analysis of the impact of HIV infection, ART, sex hormones, and inflammatory cytokines on the glycosylation machinery of tissue B cells and the activity of glycosidases in different immune cells. Our in vitro experiments with glycoengineered antibodies (depicted in Figure 9) show that manipulating the levels of IgG glycans, which are modulated in PLWH, significantly alters the anti-viral Fc-mediated innate immune functions in the predicted directions — i.e., loss of galactose decreases anti-HIV ADCC, ADCP, and ADCD. However, more investigation is essential to establish a definitive causal link between these glycomic changes and the processes of immunity, inflammation, and aging in PLWH, beyond their potential utility as biomarkers. These studies should investigate the influence of glycomic modulation on the Fc-mediated innate immune functions and inflammatory activities of antibodies in vivo, which can be accomplished through the use of animal models of HIV infection.”

- 2. The second major comment is that the authors do not explore the effects of ART alone on these parameters. Would individuals on PrEP have similar phenotypes. Are gene expression altered in these individuals for sugar enzymes? This should be explored or at the very least addressed. ART can impact the microbiome and virome and alter inflammation.*

Authors response: We agree completely with the reviewer on the importance of investigating the effects of ART alone (and different ART regimens) on our observations. Utilizing samples from pre-exposure prophylaxis (PrEP) studies would indeed be useful for these investigations. As all PLWH in our study were receiving ART, we were unable to evaluate the independent impact of HIV and ART on IgG glycomes. However, our previous studies (Ref below) (and as exemplified in **Fig. 1** on this response page) indicate that viremic infection modifies IgG N-glycans and that ART partially reverses these changes. This suggests that at least some N-glycan alterations are directly mediated by HIV infection itself.

Nevertheless, we have underscored the necessity for future research to use samples from PrEP exposed PLWoH to specifically examine the effects of ART alone on IgG N-glycans and inflammation. In particular, we stated in the Limitations section of the Discussion: “*Since our study used samples from individuals with ART-suppressed HIV infection, we cannot examine the direct impact of HIV itself in the absence of ART nor other differences between groups that could confound the results. However, our previous studies demonstrated similar alterations in IgG N-glycans in HIV viremic individuals,⁷² suggesting that at least some of the alterations observed are mediated by the infection itself. Nonetheless, further research is required to determine if ART toxicity contributes to IgG glycan alterations, and this can be investigated, at least in part, using samples from Pre-Exposure Prophylaxis (PrEP) trials. Additionally, future studies should investigate whether early initiation of ART can prevent irreversible HIV-associated N-glycan alterations.*”

- 1. Vadrevu SK, Trbojevic-Akmacic I, Kossenkov AV, Colomb F, Giron LB, Anzurez A, et al. Frontline Science: Plasma and immunoglobulin G galactosylation associate with HIV persistence during antiretroviral therapy. J Leukoc Biol. 2018;104(3):461-71. Epub 2018/04/11. doi: 10.1002/JLB.3HI1217-500R. PubMed PMID: 29633346; PubMed Central PMCID: PMC6113120.*
- 3. The third major comment is that while the authors point out and reference aberrant glycosylation has been shown in various disease states (mainly autoimmune), there is little evidence presented that IgG glycosylation is specifically associated with aging or immune age-related diseases. This should be better framed and demonstrated.*

Authors response: We have now clarified the links between altered IgG glycosylation, aging, and age-related diseases in the Introduction section. *First*, there is a robust body of literature which demonstrates that alterations to IgG glycosylation are strongly associated with both chronological and biological aging

in the general population. These alterations might even be better in predicting accelerated biological aging than traditional markers like shorter telomere length [1-5]. Such glycomic traits are significantly altered in individuals with age-related illnesses, such as cardiovascular disease (CVD) and cancer [6-8]. Second, beyond aging, extensive literature shows that IgG glycomic alterations can mechanistically influence inflammation and associated comorbidities. Three examples include: 1) Sialic acid: One of the most studied examples is intravenous immunoglobulins (IVIGs), whose anti-inflammatory activity is dependent on the sialic acid moiety carried on their Fc [9]. Beyond IVIGs, the loss of IgG sialic acid plays a mechanistic role in the development of obesity-induced hypertension, and supplementation of sialic acid reduces obesity-induced hypertension in mouse models [10]. 2) Galactose: Galactose on IgG leads to anti-inflammatory cascades by facilitating the interaction between CD32b and dectin-1 in myeloid cells [11]. 3) Fucose: Afucosylated IgG glycans contribute to inflammation during SARS-CoV-2 infection [12].

1. Kristic J, Vuckovic F, Menni C, Klaric L, Keser T, Beceheli I, et al. Glycans are a novel biomarker of chronological and biological ages. *J Gerontol A Biol Sci Med Sci*. 2014;69(7):779-89. doi: 10.1093/gerona/glt190. PubMed PMID: 24325898; PubMed Central PMCID: PMC4049143.
2. Catera M, Borelli V, Malagolini N, Chiricolo M, Venturi G, Reis CA, et al. Identification of novel plasma glycosylation-associated markers of aging. *Oncotarget*. 2016;7(7):7455-68. Epub 2016/02/04. doi: 10.18632/oncotarget.7059. PubMed PMID: 26840264; PubMed Central PMCID: PMC4884931.
3. Dall'Olio F, Vanhooren V, Chen CC, Slagboom PE, Wuhrer M, Franceschi C. N-glycomic biomarkers of biological aging and longevity: a link with inflammaging. *Ageing Res Rev*. 2013;12(2):685-98. Epub 2012/02/23. doi: 10.1016/j.arr.2012.02.002. PubMed PMID: 22353383.
4. Ruhaak LR, Uh HW, Beekman M, Koeleman CA, Hokke CH, Westendorp RG, et al. Decreased levels of bisecting GlcNAc glycoforms of IgG are associated with human longevity. *PLoS One*. 2010;5(9):e12566. Epub 2010/09/11. doi: 10.1371/journal.pone.0012566. PubMed PMID: 20830288; PubMed Central PMCID: PMC2935362.
5. Sha J, Fan J, Zhang R, Gu Y, Xu X, Ren S, et al. B-cell-specific ablation of beta-1,4-galactosyltransferase 1 prevents aging-related IgG glycans changes and improves aging phenotype in mice. *J Proteomics*. 2022;268:104717. Epub 2022/09/10. doi: 10.1016/j.jprot.2022.104717. PubMed PMID: 36084919.
6. Akinkuolie AO, Buring JE, Ridker PM, Mora S. A novel protein glycan biomarker and future cardiovascular disease events. *Journal of the American Heart Association*. 2014;3(5):e001221. Epub 2014/09/25. doi: 10.1161/JAHA.114.001221. PubMed PMID: 25249300; PubMed Central PMCID: PMC4323825.
7. Simurina M, de Haan N, Vuckovic F, Kennedy NA, Stambuk J, Falck D, et al. Glycosylation of Immunoglobulin G Associates With Clinical Features of Inflammatory Bowel Diseases. *Gastroenterology*. 2018;154(5):1320-33 e10. Epub 2018/01/09. doi: 10.1053/j.gastro.2018.01.002. PubMed PMID: 29309774; PubMed Central PMCID: PMC5880750.
8. Vuckovic F, Theodoratou E, Thaci K, Timofeeva M, Vojta A, Stambuk J, et al. IgG Glycome in Colorectal Cancer. *Clinical cancer research : an official journal of the American Association for Cancer Research*. 2016;22(12):3078-86. doi: 10.1158/1078-0432.CCR-15-1867. PubMed PMID: 26831718.
9. Kaneko Y, Nimmerjahn F, Ravetch JV. Anti-inflammatory activity of immunoglobulin G resulting from Fc sialylation. *Science*. 2006;313(5787):670-3. Epub 2006/08/05. doi: 10.1126/science.1129594. PubMed PMID: 16888140.
10. Peng J, Vongpatanasin W, Sacharidou A, Kifer D, Yuhanna IS, Banerjee S, et al. Supplementation With the Sialic Acid Precursor N-Acetyl-D-Mannosamine Breaks the Link Between Obesity and Hypertension. *Circulation*. 2019;140(24):2005-18. Epub 2019/10/11. doi: 10.1161/CIRCULATIONAHA.119.043490. PubMed PMID: 31597453; PubMed Central PMCID: PMC7027951.
11. Karsten CM, Pandey MK, Figge J, Kilchenstein R, Taylor PR, Rosas M, et al. Anti-inflammatory activity of IgG1 mediated by Fc galactosylation and association of FcγRIIB and dectin-1. *Nature medicine*. 2012;18(9):1401-6. doi: 10.1038/nm.2862. PubMed PMID: 22922409; PubMed Central PMCID: PMC3492054.
12. Chakraborty S, Gonzalez J, Edwards K, Mallajosyula V, Buzzanco AS, Sherwood R, et al. Proinflammatory IgG Fc structures in patients with severe COVID-19. *Nat Immunol*. 2021;22(1):67-73. Epub 2020/11/11. doi: 10.1038/s41590-020-00828-7. PubMed PMID: 33169014.

4. *It makes sense that IgG glycosylation could change with age as a biomarker and that autoantibodies that are IgG targeting host proteins could be impacted by glycosylation, but how would someone with HIV (not an autoimmune disease) and have aberrant glycans on IgG affect the comorbidities the authors described. For example, how would the IgG glycosylation drive cardiovascular diseases in somebody without autoimmune disease. Moreover, how would the differential glycan on IgG impact age-related illness. Most IgGs are being directed towards pathogens so it could impede pathogen response but that is not the major statement by the authors.*

Authors response: We appreciate the reviewer's insightful query, as it allows us to clarify our proposed model in the Discussion Section. Our *in vitro* data, featuring glycoengineered IgGs and depicted in **Figure 9**, shows that galactosylated IgG enhances Fc-mediated innate immune responses—such as ADCC, ADCP, and ADCD—against HIV-infected cells. Conversely, the loss of galactose on IgGs (agalactosylated IgG) diminishes these antiviral immune functions. Additionally, existing literature indicates that sialylated IgGs trigger anti-inflammatory mechanisms, likely through binding to anti-inflammatory receptors on myeloid cells. The loss of sialic acid (hypo-sialylation) on IgGs reduces the potential for these anti-inflammatory effects. Therefore, we propose a model explaining how HIV-associated agalactosylation and hypo-sialylation of IgGs may lead to increased inflammation. This proposed model, now included in the Discussion section, is further supported by **Supplementary Figure 8** (provided below). The section reads as follows:

*“In addition to serving as biomarkers for accelerated biological aging and the development of aging- and inflammation-associated diseases,[1, 13-17] IgG glycans are biologically active molecules playing significant roles in mediating immunological functions.[9, 18, 19] During ART-suppressed HIV infection, notable alterations in IgG glycosylation include the loss of galactose (agalactosylation) and sialic acid (hyposialylation). These changes are associated with increased inflammation, a higher incidence and severity of subclinical atherosclerosis, and inflammation-associated cancers in PLWH on ART. Potential mechanistic links between IgG agalactosylation or hyposialylation and increased inflammation in PLWH are illustrated in **Supplementary Fig. 8**. As shown in **Fig. 9**, IgG galactosylation enhances Fc-mediated anti-viral activities of antibodies, including ADCC, ADCP, and ADCD. Conversely, IgG agalactosylation diminishes these critical anti-viral immune functions. This reduction in anti-HIV IgG Fc-effector functions might contribute to inadequate control of virally infected cells, particularly in tissues. The effectiveness of ART in completely suppressing viral replication, especially in tissues where ART penetration may be sub-optimal, remains unclear.[20] It can be hypothesized that the compromised anti-HIV innate immune function resulting from agalactosylation could contribute to higher HIV persistence and consequently greater inflammation. This hypothesis is further supported by our previously observed negative correlations between the degree of bulk IgG galactosylation and the levels of cell-associated HIV DNA and RNA in CD4⁺ T cells during ART-suppressed HIV infection.[21]*

HIV-associated IgG agalactosylation can also directly lead to inflammation. IgG galactose facilitates interactions between FcγRIIB (CD32b) and dectin-1, triggering anti-inflammatory cascades.[11, 22, 23] Conversely, IgG agalactosylation is linked to pro-inflammatory functions by inhibiting these cascades.[11, 22, 23] Finally, sialic acid on IgGs (IgG sialylation) binds to receptors on myeloid cells, such as non-classical Fc receptors or sialic acid binding proteins (siglecs), initiating an inhibitory signal leading to anti-inflammatory responses through the inhibition of TLR4 signal transduction and consequent modulation of cytokine production.[24-29] The hypothesized effect of HIV-associated hyposialylation is to induce inflammation by reducing the potential for these anti-inflammatory cascades. Further experiments are needed to validate this model. These studies could inform the optimization of glycoengineering for broadly neutralizing antibodies in HIV cure and prevention strategies. Techniques like gene editing and metabolic glycosylation inhibition have been used to modulate antibody interactions with Fc receptors and enhance ADCC activity. Understanding specific glycomic traits impacting HIV or immune function could lead to novel glycan-based strategies to boost immune function during ART-suppressed HIV infection.”

Specific points:

1. The authors classify IgG glycans as pro-aging and anti-aging. Could they provide data with the HIV-samples that show these levels correlate with age in their cohort. There are samples from a wide age range.

Authors response: We have now included this information in **Supplementary Table 2**, which shows the correlations between each glycan trait or glycan group and age in both men and women, whether living with HIV or not. It also indicates whether the slopes of these correlations differ between people living with HIV (PLWH) and those living without HIV (PLWoH).

2. Figures and text should be appropriate PLWH and PNLWH.

Authors response: We thank the reviewer for this suggestion and have now updated all figures, tables, and text to use PLWH for people living with HIV and PLWoH for people living without HIV.

3. For figure 1 panel E, the axis should go up to 100%.

Authors response: We thank the reviewer and have fixed this oversight.

4. Figure 2, leptin is not displayed but mentioned in the text.

Authors response: In **Figure 2 A-I**, we show only the plasma markers that exhibit differences between PLWH and PLWoH in either women or men. Since leptin did not show any significant differences, it was not included in these panels. However, we have now updated **Figure 2J** to include correlations between glycans and all plasma markers, including leptin.

5. Why are there no significant differences for soluble inflammatory markers for women compared with many for men?

Authors response: This is an important observation. We were careful throughout the manuscript not to directly compare data between women and men, as women in the WIHS cohort exhibit different characteristics compared to men in the MACS cohort. In particular, the WIHS cohort consists mainly of African-American women with relatively high BMI. On the other hand, the MACS cohort is predominantly

composed of Caucasian men with relatively low BMI, as shown in **Table 1**. Therefore, we focused on comparing data from women living with HIV to their matched counterparts not living with HIV, as well as data from men living with HIV to their matched men not living with HIV, rather than directly comparing data between men and women. In these settings, several inflammatory markers differed between men living with HIV and men not living with HIV, such as IP-10, sCD163, IL-4, IL-5, IL-12p70, MIP-1 α , and MCP-2. Women living with HIV also exhibited increases in some inflammatory markers compared to women not living with HIV. Some of these markers are the same as those observed in men, such as IP-10 and sCD14, while others are unique to them, such as CXCL9. These data are detailed in Fig. 2. In other settings, these results might be different.

6. *Should there be two interaction P values one for women and men for figures 1 and 2. It appears some have an interaction that is changed by the other factors.*

Authors response: These P values in Figures 1 and 2 are derived from linear regression models assessing the interactions between HIV status (PLWH vs. PLWoH) and gender (male vs. female). Specifically, these models were examining whether the effects of HIV status differ between women and men. The P values for these interactions have been adjusted for age, BMI, and race. We have now clarified this in the text.

7. *Figures 1 and 2 should be analyzed by sex as the authors did but also by age group as age is hypothesized to affect IgG glycosylation and the levels of inflammatory markers. Then the interaction could take into account BMI and other factors. This would eliminate the effect of age and sex on these markers and allow to see the true factors that are impacted by HIV infection.*

Authors response: We have now clarified in the text that the models used for these analyses had already been adjusted for age, BMI, and race.

8. *It is unclear what the meaning or impact of 2J figure is. This should be better described in the text. It appears the data is very noisy.*

Authors response: We have now updated Figure 2J to make it simpler and highlight that the levels of IgG glycans that are enriched during HIV (agalactosylated and bisected GlcNAc) positively correlated with markers of inflammation and inflammatory aging; conversely, glycans that are depleted during HIV (galactosylated and sialylated) negatively correlated with markers of inflammatory aging.

9. *Figure 5B-C needs to be revised. A hazard model should be applied and with hazard ratios performed. Also, only the ones that vary between the HIV+ Cases and HIV- cases should be emphasized as that is the main comparison. The remaining comparisons are useful but could be provided in supplementary material to avoid confusion. Panel C is very confusing and should be revised.*

Authors response: We thank the reviewer for this valuable suggestion. We have now revised this analysis completely to include the hazard ratios for comparisons between PLWH cases and the other three groups. This analysis has made the specific glycans that differ between each of the two groups much clearer than in our original analysis.

10. *The magnitude of the differences in panes 6B-6D are concerning. It does not appear that the differences are large in magnitude. Also the p values should be displayed for panels D and E. Panels D and E should also be tested for interaction from age or BMI or other factors that could impact this finding.*

Authors response: As discussed earlier, various studies have shown that even subtle changes in glycan structures can significantly affect antibody activities. We have now included the exact P values in panels 6B and 6D. Given the small number of individuals in this analysis, constructing a model would be challenging. However, the cases and controls for this analysis were matched 1:1 for age, sex, and ethnicity, as shown in Figure 6B, which likely mitigates the effects of these variables on our findings. Nonetheless, we have emphasized the necessity for further studies with a larger cohort to address all potential confounders. This point is discussed in the revised manuscript's Discussion section.

11. Panel 7B should be removed, it does not add to the analysis.

Authors response: We agree with the reviewer and have removed this panel from Figure 7, relocating it to Supplementary Figure 6.

12. The levels of MGAT3 and FUT8 should also be displayed in 7C. With associated P values.

Authors response: We have now added these panels to Figure 7.

13. Why did the authors need to use public sequencing when they should also report the data from their CITE-seq. It is great that the public data confirm their data.

Authors response: Our own data were limited to blood samples from PLWH on suppressive ART. However, analyzing publicly available datasets allowed us to observe that the induction of genes encoding sialidase and β -galactosidase might occur in tissues such as adipose tissue. These datasets also included PLWH who are viremic. The finding that PLWH who are viremic also exhibit the induction of sialidase and β -galactosidase, as shown in **Fig. 7**, suggests that these changes are likely mediated by HIV itself, rather than ART use. These two important observations were only possible through the use of these datasets. Additionally, as the reviewer noted, they confirm our own RNAseq data.

14. There most likely are datasets of PNLWH over the lifespan, do these mRNA levels change with age naturally.

Authors response: Yes. For example, one of our main findings is that the genes expressing β -galactosidase (β -Gal, also called GLB1; the enzyme that removes galactose from proteins) and neuraminidase (sialidase, also called NEU1; the enzyme that removes sialic acid from proteins) are elevated in cells and tissues from PLWH compared to those without HIV. We have confirmed that the protein levels of β -galactosidase are also higher in the plasma of PLWH than in PLWoH, and this elevation correlates negatively with the levels of galactose on IgG glycans.

β -galactosidase (" β -Gal") is a well-established marker of cellular senescence. Reviewing the latest literature, we identified a recent study using a dataset that investigated, at a single-cell level, the relationship between gene expression in lungs and aging in humans. Interestingly, both GLB1 and NEU1 exhibited strong positive correlations with age in both mesenchymal and immune cells (FDR $p < 0.001$).

1. De Man, R. et al. A Multi-omic Analysis of the Human Lung Reveals Distinct Cell Specific Aging and Senescence Molecular Programs. bioRxiv, doi:10.1101/2023.04.19.536722 (2023).

These data further suggest that these glycan-modifying enzymes are induced in PLWH, despite ART, also linked to chronological aging. We have now highlighted this in the Discussion section and emphasized the need for more comprehensive analyses of the shared molecular alterations between chronological aging and the potential biological aging caused by living with a chronic viral infection.

Reviewer #2

Reviewer general comment: *The manuscript “Plasma Glycomic Markers of Accelerated Biological Aging During Chronic HIV Infection” by Giron et al brings valuable new data about IgG glycosylation in HIV-positive individuals on anti-retroviral therapy. This study is worth publishing in Nature Communications assuming the following major and minor comments are properly addressed.*

Authors response: We thank the reviewer for their positive views on our study. We have now addressed all major and minor comments by conducting new experiments to evaluate any potential contamination in our isolated IgGs, performing new analyses including rerunning our machine learning models with and without the A2 (now called A2G2S2) glycan, and implementing several other modifications throughout the manuscript, as detailed below.

Major comments:

1. Ln232-241: LASSO model selected G0, G0FB, G1FB, and A2 glycans as the best predictor of chronological age. This aligns well with published data, but there is a caveat that should be checked. A2 glycan is a very minor component of the IgG glycome, while it is the major component of the transferrin glycome (PMID: 36959410). Since transferrin is the major contaminant of IgG during affinity capture, A2 glycan is often considered to be an indicator of contamination and it should not be assumed that it originates from IgG without additional confirmation that transferrin contamination was significantly below 1% (which is not easy to achieve). I believe G2F glycan may be ranked just a bit lower than A2 in LASSO selection, and excluding A2 and adding G2F to the model may be an easy solution.

Authors response: We thank the reviewer for raising this important concern. To address it, we implemented two key strategies:

1. **Analysis of the Purity of Our Isolated IgGs:** We conducted further examination of randomly selected IgGs isolated in this study through: **a)** SDS-PAGE, as illustrated in **Supplementary Figure 1C**. The data from this analysis show high purity levels, comparable to commercial IgG samples from Sigma Aldrich. **b)** Liquid Chromatography-Tandem Mass Spectrometry (LC-MS/MS) analysis, as depicted in **Supplementary Figure 1D**. This method augmented the previous analysis, reaffirming the high purity of our IgG preparations and revealing an extremely low level of Transferrin contamination (0.001%).
2. **Reevaluation of Machine Learning Models:** Acknowledging the possibility of contamination in any of the 1,808 samples analyzed, and in line with the reviewer’s suggestion, we revised our machine learning models. This involved excluding the A2 (now called A2G2S2) glycan and incorporating only the remaining three glycan structures into our model. The revised model, presented in **Figure 4B**, yielded data consistent with the original model (in **Figure 4A**) calculated using all four glycan structures.

These combined efforts — analyzing IgG purity and recalibrating our models with both three and four glycan structures and obtaining similar results — strengthen our confidence in the rigor and reproducibility of our reported findings.

2. In Figure 3 Trait A1 is labelled as pro-ageing. Published data consistently show that sialylation decreases with age, thus this trait should be anti-ageing. Also, non-core fucosylated monosialylated glycans originate more from contaminating other proteins than from IgG, thus utmost care to estimate levels of contaminating proteins is needed before any claims about this structure can be made, in particular when the results go against previously published data.

Author's Response: We acknowledge the reviewer's comments and have removed the A1 trait from Figure 3. It is possible that specific glycans correlate with aging differently, depending on the context, especially given the specific population we are investigating in this manuscript. In order to provide a complete set of data, we are now including **Supplementary Table 2**, which shows the correlations between each glycan trait or glycan group and chronological age in both men and women, regardless of HIV status. This table also indicates whether the slopes of these correlations differ between people

living with HIV (PLWH) and those living without HIV (PLWoH). For details countering the question of whether the findings are due to contamination, please refer to our comments above and the new data presented in **Supplementary Figures 1C and 1D**.

3. *From the description in Ins 274-277 it is not clear which glycans were analysed in this part of the study. The first sentence refers to “glycomic dysregulation linked with inflammation (Fig 3)”, but Fig 3 is showing age-related changes. Among the major IgG glycans the most extensive association with HIV+ART+ status was the extensive decrease in G1F[3] in women, which was recently identified to be the best predictor of future CVD events specifically in women (PMID 31915204 and 36174116). This glycan is not named here, but it is not clear whether it was not significant, or perhaps not analysed at all (i.e. grouped in G1F together with G1F[6]). Small number of women in this part of the study may also be the reason why it was not identified, but this should be clarified and perhaps also mentioned in the discussion.*

Author's Response: We thank the reviewer and have clarified this section accordingly. The new analysis presented in **Figure 5** demonstrates that G1F[3] (now renamed FA2[3]G1) is indeed a significant glycan trait associated with CVD events in our dataset. This finding is consistent with the literature the reviewer referred to. We have now cited these studies to further corroborate our findings.

Minor comments:

1. *Naming of glycans used in this paper follows commercial Luder nomenclature that is somewhat confusing in using A1 and A2 for mono and disialylated glycans. “Oxford notation” would be better to avoid confusion.*

Authors response: We thank the reviewer for this suggestion. We have now renamed all glycans in the manuscript, figures, and tables to follow the Oxford notation.

2. *Section introducing functional aspects of IgG glycosylation would benefit from inclusion of recent data reviewed in PMID 37414906.*

Authors response: We thank the reviewer for directing our attention to this review article. We have revised the sections that discuss the functional significance of IgG glycans, incorporating references from this review and other sources.

3. *LN 501-502: Authors implicate that the change in expression of glycosyltransferases may be the causative element controlling IgG glycosylation. However, a number of studies demonstrated limited correlation between glycosyltransferase expression and glycome composition. Furthermore, GWAS studies indicated that glycosyltransferases are only a small part of the regulatory network that control glycosylation (PMID 32128391).*

Authors response: We agree with the reviewer. In fact, our data also demonstrated a limited correlation between the expression of glycosyltransferases in B cells and IgG glycans. However, the strongest correlation with IgG glycans that we observed was with the levels of glycosidases, particularly neuraminidase and β -galactosidase, especially those circulating in the plasma and potentially expressed by various cells and tissues. Nevertheless, we have updated this section in the Discussion to highlight the findings from various GWAS studies. We emphasize that the determinants of IgG glycosylation are likely multifactorial and involve the expression of several genes. Some of these genes might be directly related to the glycosylation machinery, while others may not.

Reviewer #3

Reviewer general comment: In this study, the authors seek to profile the N-glycans of plasma IgG in a large group of HIV patients undergoing anti-viral treatment and in a matching HIV negative control cohort not receiving treatment. The focus is to study the accelerated aging experienced by HIV patients receiving treatment; the rationale to focus on IgG N-glycosylation as a proxy for aging is that others have previously linked the IgG N-glycans in plasma to biological aging. Profiling and quantitation of the released and labelled N-glycans were performed using high throughput CE from which several comparative analyses of the IgG glycans were performed after grouping the individual N-glycans for terminal traits (galactose, agalactose, fucose, bisecting GlcNAc termini etc). Several other types of experiments were performed to follow up on the initial results from the IgG glycoprofiling including measurements of inflammatory markers, glycosylation enzymes and cytotoxicity on both the patient IgGs and also glycoengineered IgG species. Given the importance of the glycan profiling data of IgG from the donor groups (Figure 1-8, Table 2-3) and from the glycoengineered IgG (Figure 9) I have selectively focused on providing comments to this part which fits with my expertise profile.

Authors response: We thank the reviewer and have addressed all concerns with new experiments to evaluate the potential contamination in our isolated IgGs. We have also assessed whether this isolation impacted the percentage of IgG1-4. Additionally, we have added a detailed description in our Methods section for isolating IgG and examining their N-linked glycans, as detailed below.

Major comments:

- 1) *Data transparency and data sharing:* Given the critical reliance on the accuracy of the IgG glycan identification and quantitation for the conclusions of this work, all data (raw, processed and annotated/interpreted) must be made available to the community via publicly accessible repositories to allow for scrutiny and further interrogation by others as per good data sharing practices and conventions in the field. This applies to all CE and MS glycomics and glycopeptide data (and presumably also to the single cell transcriptomics data albeit not covered in my review comments). The text in the current Data Availability section is not appropriate as it does not allow for the expected data transparency and sharing.

Authors response: We thank the reviewer, and we have deposited all data into different repositories:

1. Mass spectrometry proteomic data were deposited to **ProteomeXchange** via the PRIDE database.

Project accession: PXD046510
Project DOI: 10.6019/PXD046510
Username: reviewer_pxd046510@ebi.ac.uk
Password: QgSrNFYz

2. CE data: while we are not aware of public repositories for CE data, we have uploaded a full dataset containing the data for 1,808 samples in **Zenodo** and plan to make this public upon the publication of this manuscript. Currently, this dataset can be viewed by reviewers using this link:

https://zenodo.org/records/10553465?token=eyJhbGciOiJIUzUxMiJ9.eyJpZCI6ImQxMGYyZDc3LTZjOWEtNDk4OC05YWYxLTgyZTVlOTUyZWNiNSIsImRhdGEiOnt9LjYyZW5kb20iOiIzNWY3ZTBhNzI4MzVmYTUwOTIhZWZkYmNhMjE1N2JjZCJ9.2dd0xY9p6ZPMFyOS73Rz11haGXlYObjCtKmlrawiTh9aqrRV1dW5xv4hwp68IV_RTELFwnSTwSGMxius1I8zQ

2. *Glycan identification and profiling:* The equally critical method section of the IgG N-glycan analysis is weak and does neither allow the reader to fully understand each experimental step of the glycan analysis let alone reproduce the experiments, which are minimum expectations given the importance of the IgG glycan profiling for the conclusions of the study. As an analyst, it is still unclear to me how a confident identification of each of the N-glycans was performed in this study; **were retention time**

standards applied and how were they used to produce confident glycan structure identification down to the linkage/branching level? Were RT libraries generated or already available? Looking at SFig S1, one peak is labelled B334.99; this peak is also included in Table 2, but labelled as an undefined trait. Why is this undefined peak included in the glycan profile? Another major peak at 220 migration units is consistently abundant in all traces, but was neither assigned nor included in the N-glycan profile. Without analytical evidence and a detailed explanation of the analytical approach this reviewer is not convinced about the identities of the assigned peaks.

Authors response: We thank the reviewer and have now updated the IgG N-glycan analysis section in the Methods to include all the requested details. This section now reads:

“IgG N-glycan analysis. IgG N-glycan analyses were performed using the GlycanAssure APTS Kit (Thermo Fisher, catalog # A33952), following the manufacturer's protocol. Specifically, IgG samples were incubated with the denaturing reagents for 5 minutes at 80°C. After denaturing, samples were digested with PNGase enzyme for 10 minutes at 50°C. Post-digestion, released glycans were labeled using APTS reagent and incubated for 60 minutes at 50°C. The reaction cleanup was performed using positive selection with magnetic beads provided in the kit. To prepare the run, labeled and purified glycans were spiked in with GeneScan 600 LIZ size standard (Thermo Fisher; catalog #4408399) to ensure consistent peak heights and to provide consistent results between different injections and capillaries. The N-glycans were analyzed using the 3500 Genetic Analyzer capillary electrophoresis system, as detailed in a previous publication:

- Xu Z, Ho M, Bordoloi D, et al. Techniques for Developing and Assessing Immune Responses Induced by Synthetic DNA Vaccines for Emerging Infectious Diseases. *Methods Mol Biol.* 2022;2410:229-263. doi: 10.1007/978-1-0716-1884-4_11. PMID: 34914050.

Each sample was run in a single well, and samples were added to the machine in batches of 96 samples in a random order. Run parameters were: oven temperature at 60°C, run time of 1330 seconds, injection time of 24 seconds, injection voltage of 1.6 kVolts, and run voltage of 19.5 kVolts. Identification of the glycan structures from relative migration units (RMU) was based on commercially available known glycan traits labeled with APTS evaluated using the same system by Thermo Fisher and others in several published and commercially available protocols (see below). The relative abundance of N-glycan structures was quantified by calculating the area under the curve of each glycan structure divided by the total area under the curve from all glycan peaks using the Applied Biosystems GlycanAssure Data Analysis Software Version 2.0.

- Cajic S, Hennig R, Burock R, Rapp E. Capillary (Gel) Electrophoresis-Based Methods for Immunoglobulin (G) Glycosylation Analysis. *Exp Suppl.* 2021;112:137-72. doi: 10.1007/978-3-030-76912-3_4. PubMed PMID: 34687009.
- Feng HT, Li P, Rui G, et al. Multiplexing N-glycan analysis by DNA analyzer. *Electrophoresis.* 2017;38(13-14):1788-99. doi: 10.1002/elps.201600404. PubMed PMID: 28426178.
- Feng HT, Su M, Rifai FN, et al. Parallel analysis and orthogonal identification of N-glycans with different capillary electrophoresis mechanisms. *Anal Chim Acta.* 2017;953:79-86. doi: 10.1016/j.aca.2016.11.043. PubMed PMID: 28010746.
- Mahan AE, Tedesco J, Dionne K, et al. A method for high-throughput, sensitive analysis of IgG Fc and Fab glycosylation by capillary electrophoresis. *J Immunol Methods.* 2015;417:34-44. doi: 10.1016/j.jim.2014.12.004. PubMed PMID: 25523925; PubMed Central PMCID: PMC5054724.
- Lu LL, Das J, Grace PS, Fortune SM, Restrepo BI, Alter G. Antibody Fc Glycosylation Discriminates Between Latent and Active Tuberculosis. *J Infect Dis.* 2020;222(12):2093-102. Epub 2020/02/16. doi: 10.1093/infdis/jiz643. PubMed PMID: 32060529; PubMed Central PMCID: PMC5054724.

The identities of the peaks were based on already available libraries in the Applied Biosystems GlycanAssure Version 2.0 software. The software calculates a normalization factor based on a threshold setting. For each injection, the normalization factor is used as a multiplier to adjust the peak

height of the sample peaks relative to the GS600 LIZ size standard peaks. The normalization factor has minimum and maximum limits, so if the size standard peak heights are abnormally high or low, the normalization will be limited. Peaks are then assigned to a glycan structure based on RMU, and the area under the curve is calculated automatically by the software.

Several blank controls (composed of GeneScan 600 LIZ size standard, Landmark Red, and CE loading buffer) were run periodically. All peaks were assigned and identified, except for two peaks: a peak around RMU of 220, determined as a carryover based on blank samples (**Supplementary Fig. 1A**) and hence was not assigned; and another peak at around RMU of 334, which was assigned but not identified as different available libraries from Thermo Fisher and others (below) have conflicted information on this peak's identity - some label it as A2G2, others as FA2BG1, and others do not assign it. Therefore, to ensure rigor and reproducibility of our results, we elected to assign it but not identify it.

- Cajic S, Hennig R, Burock R, et al. Capillary (Gel) Electrophoresis-Based Methods for Immunoglobulin (G) Glycosylation Analysis. *Exp Suppl.* 2021;112:137-72. doi: 10.1007/978-3-030-76912-3_4. PubMed PMID: 34687009.
- Grace PS, Dolatshahi S, Lu LL, et al. Antibody Subclass and Glycosylation Shift Following Effective TB Treatment. *Front Immunol.* 2021;12:679973. doi: 10.3389/fimmu.2021.679973. PubMed PMID: 34290702; PubMed Central PMCID: PMC8287567.
- Feng HT, Li P, Rui G, et al. Multiplexing N-glycan analysis by DNA analyzer. *Electrophoresis.* 2017;38(13-14):1788-99. doi: 10.1002/elps.201600404. PubMed PMID: 28426178.
- Mahan AE, Tedesco J, Dionne K, et al. A method for high-throughput, sensitive analysis of IgG Fc and Fab glycosylation by capillary electrophoresis. *J Immunol Methods.* 2015;417:34-44. doi: 10.1016/j.jim.2014.12.004. PubMed PMID: 25523925; PubMed Central PMCID: PMC5054724.

3. *Glycan quantitation and cohort acquisition: An important aspect of this study is that 1200+ samples were quantitatively profiled for IgG glycosylation which is not a trivial analytical feat; if not performed appropriately, such large scale data generation could lead to systemic bias in the observed IgG glycan levels across sample sets. The manuscript does not discuss any analytical aspects or issues relating to this point. It should be detailed exactly how the sample cohort was acquired (e.g. **were samples scrambled in their injection order and run in one single batch or were individual patient groups analyzed one by one across multiple batches?**). **Were each of the samples analyzed once or in multiple technical replicates? Were standards run between samples to calibrate for retention time. What were the CE settings? Were blanks run to rule out carry over? Were the baselines and AUCs manually determined or performed by a software?** These details must be detailed in the method section and the discussion should discuss any potential analytical weaknesses and unintended biases that could be introduced by the experimental design and any efforts to mitigate such unwanted effects.*

Authors' Response: Please refer to our detailed methods in the response to the second major concern, where we describe how the samples were run, our assessment of blank samples to rule out carryover, and the software used in the analysis.

4. *Sample preparation: The IgG N-glycan profiling relies blindly on the ability of a Protein G column to quantitatively and consistently capture all IgG glycoforms (and only IgG) before PNGase F digestion and profiling. 50 ul of plasma (in excess of 1 mg IgG) was used for the capture which is a high amount and may be pushing towards the capacity limit of the column if not careful? Can it be ruled out that some of the differences in IgG glycosylation observed across patient groups is in fact introduced at the protein G column stage such as i) co-purification of other glycoproteins, ii) IgG subtype switching i.e. IgG1 towards IgG2-4 or iii) a selective capture of some IgG glycoforms over others? **The authors should demonstrate that the protein G column captures only and repeatedly IgG under the applied***

conditions since this is the only way to ensure the glycan data is in fact originating from plasma IgG. The corresponding method section should be expanded to detail the protein G isolation procedure (e.g. was the same protein G column used for multiple samples and, if so, how was carryover avoided?) and the deglycosylation and labelling reactions should be explicitly explained. Deglycosylation was presumably not performed of natively folded IgG as indicated in the current method section and how was the excess labelling reagent and the deglycosylated protein removed from the labelled glycans before CE analysis?

Authors' Response: We thank the reviewer, and we have now updated the Methods section on isolating IgGs as follows:

IgG Isolation. IgG was purified from 50 µl of plasma using the Pierce Protein G Spin Plate (Thermo Fisher, catalog #45204). Each well of the isolation plate contained 50 µl of resin, capable of purifying IgGs from up to 100 µl of serum. The protein G resin plate was discarded after each single use to avoid carryover and cross-contamination between samples. IgG was quantified using a BCA kit (Thermo Fisher, catalog # 23225) with bovine gamma globulin standard (Thermo Fisher, catalog# 23213). To confirm the purity of the IgG isolated using this method, we ran four random isolated IgG samples alongside a commercially available pure IgG sample from Sigma Aldrich (Catalog #I2511) on an SDS-PAGE and estimated the purity using densitometry analysis. High purity of the isolated IgG was observed, as shown in **Supplementary Figure 1C**. For this analysis, IgGs were run under denaturing conditions using Bolt Bis-Tris Plus Mini Protein Gels, 4 - 12%, 1.0 mm (Invitrogen, catalog # NW04120BOX). The gel was run for 22 minutes at 220 volts. The gel was then washed three times for 5 minutes each with distilled water, followed by staining with Simple Blue (Invitrogen, catalog number LC6065) for 1 hour.

To further confirm the purity of isolated IgGs using this method, we used Liquid Chromatography-Tandem Mass Spectrometry (LC-MS/MS) analysis on a random IgG sample along with the commercial IgG from Sigma Aldrich. As depicted in **Supplementary Figure 1D**, this method confirmed the SDS-PAGE analysis, reaffirming the high purity of our IgG preparations and revealing a low-level contamination with other proteins. For this analysis, samples were electrophoresed into Bolt Bis-Tris Plus Mini Protein Gels, 4 - 12%, 1.0 mm (Invitrogen, catalog # NW04120BOX), and stained with Coomassie blue. The entire stained gel regions were excised, and in-gel digested with trypsin for LC-MS/MS analysis using a Q Exactive Plus mass spectrometer (Thermo Fisher Scientific) coupled to a Vanquish Neo UHPLC system (Thermo Fisher Scientific). MS data were searched against a UniProt human protein database (August 2023) and a common contaminant database using MaxQuant version 2.4.7.0 (Ref: PMID 19029910). Consensus identification lists were generated with false discovery rates set at 1% for protein and peptide identifications.

Finally, to examine whether our IgG isolation method significantly impacted the percentage of different IgG subclasses, we analyzed the IgG subclasses in six paired plasma samples and isolated IgG from these samples using ELISA. As shown in **Supplementary Figure 1E**, isolated IgGs contained the expected levels of IgG1 (approximately 60-70% of total IgGs), similar to the plasma, and about 20-30% IgG2-4, again similar to the plasma. For this analysis, levels of IgG1, IgG2, IgG3, and IgG4 were quantified using the IgG Subclass Human ELISA Kit (Invitrogen, Catalog# 99-1000) according to the manufacturer's instructions."

For questions regarding deglycosylation and cleaning up steps in CE analysis, please refer to our detailed methods in the response to the second major concern, where we describe these steps.

Minor comments:

1. *Title and Short Title, P1: Instead of "plasma glycomic markers" the title should for accuracy be changed to "IgG glycan markers". The manuscript repeatedly uses "glycomics" and "IgG glycomics/glycome".*

The -omics/-ome suffix should by convention be reserved for system-wide analyses which is not performed in this study (IgG centric). Use instead IgG N-glycans or IgG N-glycosylation.

Authors response: We thank the reviewer and have now updated the titles and text to use 'IgG glycans' and 'IgG N-glycans' instead of 'glycomics/glycome’.

2. *Abstract, P1: The manuscript is difficult to read due to the many abbreviations, consider if some abbreviations are unnecessary such as PWH (seems redundant given that MWH and WWH are also in use).*

Authors response: We have tried to limit the use of abbreviations throughout the manuscript to enhance its readability.

3. *Introduction: Several references are quite dated (up to 10-15 years old, e.g. some amongst these references Ref6-9, 10-11, 12-14, 15-17, 26-27. Can more recent literature be cited?*

Authors response: We have now updated these sections and their cited literature.

4. *L110 and L116 (and in Table 1): Are these cell counts measured in blood?*

Authors response: Yes, we have now made this clarification in both the text and Table 1.

5. *L117: Should be 20 individual N-glycan structures since B334.99 is not a glycan until proven otherwise (see comment above).*

Authors response: That is correct, and we have now updated this sentence.

6. *L157: Figure 1G: Given the relatively subtle changes observed in the IgG glycosylation in HIV patients the indicated glycan transformation is hugely exaggerated which should be pointed out to uninformed readers.*

Authors response: We thank the reviewer for this comment, and we have now pointed this out in the Discussion section.

7. *L159 and L187: What is meant by “higher markers” and “higher inflammation”? Systemic markers/inflammation better?*

Authors response: We agree with the reviewer and have updated these sentences as suggested.

8. *L169: leptin with lowercase.*

Authors response: We have now fixed this.

9. *L247: “no inflammatory markers alone correlated....”?*

Authors response: We have now fixed this.

10. *L285 and elsewhere: anti- and proinflammatory glycans. Better with glycans associated with anti-or proinflammatory conditions to avoid implying causative relationships?*

Authors response: We agree with the reviewer and have now fixed this.

11. *L336: “...observed HIV/ART-mediated glycomic alterations” change to “...IgG N-glycan alterations associated with HIV/ART” to avoid implying direct/causative relationship? Same for L449.*

Authors response: We have rephrased both sentences.

12. *L346: Unbiased to unsupervised?*

Authors response: We have now changed unbiased to unsupervised.

13. *L625: Why was both Byonic and pGlyco used? Only Byonic was detailed below.*

Authors response: We apologize for this oversight and have deleted pGlyco from the Methods.

14. L633: 100ug of the material. What is meant by material? Purified IgG?

Authors response: Yes, IgGs. We have now made this clearer in the text.

15. Figure 1: Do the number of individuals in Figure 1 add up to match 1,216 individuals mentioned in the abstract?

Authors' response: The total number of samples analyzed in this study was 1,808, belonging to 1,214 unique individuals. The breakdown of these samples is as follows:

- 985 cross-sectional samples from women and men, living with or without HIV; the results of these samples are presented in Figures 1, 2, 3, 4, 7, and 8.
- 622 longitudinal samples from 94 individuals, living with or without HIV, with data presented in Supplementary Figure 4.
- 112 cross-sectional samples from people living with or without HIV, and with or without CVD, analyzed in the context of the analysis presented in Figure 5.
- Finally, 89 longitudinal samples from 23 individuals (10 cases and 13 controls) analyzed in the context of the data in Figure 7.

Now we have uploaded all data to Zenodo; this became clearer. We have now corrected this in the manuscript.

16. All figures: The core fucose in all depicted glycan structures should by convention be changed to facing the right-hand side of the N-glycan since this is an alpha1-6 linked (not alpha1-3) fucose. Supplementary Figure S1: The y axis looks like absolute (not relative) FUs, please check.

Authors response: We thank the reviewer, and we have now fixed orientation of the fucose across all figures. The y-axis in Supplementary Figure 1 is relative migration units (RMU).

REVIEWERS' COMMENTS

Reviewer #1 (Remarks to the Author):

The authors have addressed the reviewers concerns and as a result improved the manuscript. I have no further issues.

Reviewer #3 (Remarks to the Author):

The authors have made a good job responding to the raised concerns which has considerably improved the quality of the manuscript. Before acceptance I request that all core fucose residues are depicted correctly when drawing the glycan cartoons (pointing right) in Figure 7 and sFig S8B-C (please check all figures).

Reviewer #3

Reviewer comment: The authors have made a good job responding to the raised concerns which has considerably improved the quality of the manuscript. Before acceptance I request that all core fucose residues are depicted correctly when drawing the glycan cartoons (pointing right) in Figure 7 and sFig S8B-C (please check all figures).

Authors response: We thank the reviewer and apologize for this oversight and have now fixed these two figures.